# Effects of Submarine Groundwater on Nutrient Concentration and Primary Production in a Deep Bay of the Japan Sea

Menghong Dong[1], Xinyu Guo[2*], Takuya Matsuura[3], Taichi Tebakari[4], and Jing Zhang[5]

[1]State Key Laboratory of Satellite Ocean Environment Dynamics, Second Institute of Oceanography, Ministry of Natural Resources, Hangzhou, China
[2]Center for Marine Environmental Studies, Ehime University, Matsuyama, Japan
[3]Graduate Faculty of Interdisciplinary Research Faculty of Engineering, Civil Engineering and Environmental Engineering, Yamanashi University, Koufu, Japan
[4]Civil, Human and Environmental Science and Engineering Course, Graduate School of Science and Engineering, Chuo University, Tokyo, Japan
[5]Faculty of Science, Academic Assembly, University of Toyama, Toyama, Japan

*Correspondence to*: Xinyu Guo (guo.xinyu.mz@ehime-u.ac.jp), ORCID ID: 0000-0002-4832-8625

**Abstract.** We constructed a coupled physical-ecosystem model with a tracking module to evaluate the influence of submarine groundwater discharge (SGD) and river water on nutrient distribution and phytoplankton growth in Toyama Bay, a deep bay in the Japan Sea. The tracking technique allows us to distinguish SGD- and river-derived nutrients in the bay and evaluate their contributions to the nutrient inventory and phytoplankton growth. Horizontally, SGD-derived nutrients were primarily distributed within a narrow band from the coastline (< 3 km), and vertically, they were abundant in the middle and bottom layers (>5 m depth). Because of the buoyancy of SGD, SGD-derived nutrients were transported upward to the surface layer and used by the phytoplankton for growth. The contribution of SGD-derived nutrients to phytoplankton growth within a narrow band from the coastline is highest from June to August, exceeding 10%, with an annual average of 4%. On the other hand, river water exerted a greater effect on phytoplankton growth than SGD did, on both the spatial range and the amount of phytoplankton biomass. Due to the different distributions of river- and SGD-derived nutrients, their proportions used by phytoplankton differed from coastal to offshore areas. These findings enhance our understanding of the coastal ecosystems affected by land water.

## 1 Introduction

Submarine groundwater discharge (SGD) is a process that involves the release of groundwater from the seafloor into a coastal sea (Burnett et al., 2003; Moore, 2010). Based on the monitoring of radioactive elements such as radium and radon, the occurrence of SGD in coastal seas is well established. It was also suggested that SGD is a pathway by which freshwater and nutrients from the terrestrial groundwater system enter the marine environment (Hatta & Zhang, 2013; Lecher & Mackey, 2018; Luijendijk et al., 2020; Moore, 1996; Müller et al., 2023; Santos et al., 2021; Wang et al., 2018). The global amount of SGD is estimated to be as high as 6–7% of total surface water input to the world ocean system (Zektser, 2000) or as high as 10% of river discharge in the world ocean system (Burnett et al., 2001; Taniguchi et al., 2002).

Owing to the high nutrient concentration of groundwater, it was suggested that SGD is accompanied by the release of large amounts of nutrients into the coastal sea (Burnett et al., 2001, 2003; Church, 1996; Rodellas et al., 2015; Taniguchi et al., 2002; Wang et al., 2014; Zhang & Satake, 2003). Local- and global-scale assessments of SGD suggested that in coastal areas, SGD-derived nutrient inputs are comparable to or even higher than river-derived nutrient loads (Hatta et al., 2005; Cho et al., 2018; Zhang et al., 2020). Such significant SGD inputs can influence nutrient budgets and initiate phytoplankton blooms in the sea (Lapointe et al., 1990). Based on high Chlorophyll concentrations at sites with high SGD, Santos et al. (2021), in a report that summarized the impact of SGD-related nutrients on marine ecosystems, noted that the most documented response to SGD-

derived nutrients is an increase in phytoplankton growth (Hwang et al., 2005; Taylor et al., 2006). Nutrients exported via SGD are critical to coastal biogeochemistry and act as a significant nutrient source to the oceans (Wilson et al., 2024).

Two major concerns are associated with the interpretation of SGD-related observations. First, confirming the distribution of SGD-derived nutrients in the coastal seas is challenging, even though we know these nutrients are released at the sea bottom. In coastal seas, in addition to SGD-derived nutrients, river water also inputs nutrients. To distinguish the nutrients from different sources is challenging in the observations as river water and SGD eventually mix. The second one is that it remains unclear how much of the SGD-derived nutrients is consumed by phytoplankton. In principle, SGD-derived nutrients influence phytoplankton growth only after reaching the euphotic layer. Most previous observations on SGD were carried out on the seas with a water depth < 10 m. In such shallow seas, owing to wind- or tide-induced mixing, SGD-derived nutrients are considered to easily reach the surface layer where there is sufficient sunlight (Bratton, 2010; Jiao and Post, 2019; Müller et al., 2023). However, it was also reported that sometimes drastic changes in the water column above SGD sites also make it difficult to confirm the response of phytoplankton growth to SGD-derived nutrients (Sugimoto et al., 2017). On the other hand, for the SGD released at the deep seafloor, the effects of wind- and tide-induced mixing are limited and the sunlight required for phytoplankton growth at depths where the SGD can reach may be insufficient for primary production (Cloern et al., 2014). Therefore, the second concern is particularly serious for the deep coastal seas.

The interest in knowing the effects of different origins of nutrients in the marine ecosystem is not limited to SGD. As a useful tool, the implementation of a tracking module in the ecosystem simulation to track nutrients from different sources has been proposed (Kawamiya, 2001; Ménesguen et al., 2006). This technique has been successfully applied in the Baltic Sea (Neumann, 2007), the Japan Sea (Onitsuka et al., 2007), the Northwest Atlantic areas (Lenhart and Große, 2018; Lenhart et al., 2013; Painting et al., 2013), the East China Sea (Zhang et al., 2019), and the Seto Inland Sea (Leng et al., 2023). Thus, a comprehensive evaluation of SGD and its impacts on the marine ecosystem, which is urgently required, is possible by applying such a numerical ecosystem model with a tracking module. Comparing the contributions of SGD-derived nutrients with those of nutrients from other landside-derived sources such as river water to phytoplankton growth will enhance understanding regarding the integrated land-sea nutrient cycle, ecosystem, and hence eutrophication issues.

Here, we selected Toyama Bay, a deep bay in the Japan Sea (Figure 1a), as our study area. This bay is characterized by a narrow continental shelf and steep slopes, with a trough across the continual shelf and a slope toward the inner part of the bay (Figure 1b). On the landside of the bay, there are many mountains up to 3,000 m in height. These mountains are characterized by steep slopes from the mountains to the coast. Further, large amounts of freshwater flow into this bay during the snowmelt season (Watanabe et al., 2013). This bay is ideal for studying SGD because it is easily accessible and has several SGD sites off the eastern coastal area. These SGD sites are distributed from the shore to relatively deep waters (Hatta & Zhang, 2013; Koyama et al., 2005; Nakaguchi et al., 2005; Zhang et al., 2005; Zhang & Satake, 2003). The reported SGD flow rates in the eastern coastal area of the bay, observed as approximately $72$–$187 \ cm \ day^{-1}$ (Zhang et al., 2005), are greater than the global average of $6.5 \ cm \ day^{-1}$ (Santos et al., 2021). Specifically, the SGD in this area is estimated to be ~13–20% of the entire river water discharge into the bay (Hatta et al., 2005; Hatta and Zhang, 2013). The dissolved inorganic nitrogen (DIN) and dissolved inorganic phosphorus (DIP) fluxes entering Toyama Bay via SGD, estimated using a box model, are 2.13 $mmol \ m^{-2} \ per \ day$ and 0.02 $mmol \ m^{-2} \ per \ day$, respectively (Hatta et al., 2005), which are comparable to the global averages of 4.06 $mmol \ m^{-2} \ per \ day$ and 0.06 $mmol \ m^{-2} \ per \ day$ (Santos et al., 2021).

Additionally, Toyama Bay is known for its high biological productivity and unique oceanographic features, which support commercially important fishery resources. While no severe eutrophication issues have been reported (Tsujimoto, 2012), nutrient dynamics play a crucial role in shaping primary production, which directly impacts the ecosystem and fisheries in the bay (Northwest Pacific Region Environmental Cooperation Center, 2010). Given the recent reduction in nutrient inputs from riverine sources due to improved wastewater treatment (Katazakai and Zhang, 2021a) and the decrease in nutrient concentrations in SGD caused by climate change-induced shifts from snowfall to rainfall in midlatitude Japan (Katazakai and

Zhang, 2021b), understanding the role of SGD as a nutrient source is essential for predicting future changes in primary production.

To investigate the spatiotemporal distribution of SGD-derived nutrients and clarify their contribution to the phytoplankton growth in the bay, we developed a coupled physical-ecosystem model that covers the entire Toyama Bay area and considered SGD and riverine nutrient inputs. We used the numerical model to investigate the nutrient dynamics and marine ecosystem structure of this bay. Further, using the tracking technique implemented in the model, we examined the distribution of SGD-derived nutrients and nutrients from other sources and evaluated their contributions to the nutrient inventory and phytoplankton growth. Finally, we examined the importance of the buoyancy effect of SGD in controlling the upward transport of SGD-derived nutrients.

## 2 Model configuration

### 2.1 Hydrodynamic model

In this study, we used the Stony Brook Parallel Ocean Model (sbPOM), a parallel version of the Princeton Ocean Model as the hydrodynamic model. Some additional information on this model is provided in Text S1. The model domain covers an area of approximately 150 km × 150 km including Toyama Bay (Figure 1b). Given that the inner part of the bay is considerably affected by freshwater input and the sizes of the river mouths along the coast are relatively small, we used a variable grid system with a fine resolution in the inner part and a coarse resolution in the outer part. In the inner part (area (A) in Figure 1b), the grid size is 10 seconds (~300 m) in the zonal and meridional directions. In the other three areas, the grid size is 30 seconds (zonal) x 10 seconds (meridional) in area (B), 10 seconds (zonal) x 30 seconds (meridional) in area (C), and 30 seconds (both zonal and meridional) in area (D). To avoid an abrupt change in grid size, we set a transition zone of ~5.5 km between the areas of 10'' and 30''. In the vertical direction, there are 31 sigma layers, and the vertical level range is fine near the surface layer. The topography used in the model is based on the 30-second topography data (JTOPO30) prepared by the Marine Information Research Center (MIRC), Japan Hydrographic Association. Two modifications were made to the original topography data: setting the minimum (5 m) depth in the model and smoothing the topography data to satisfy the following criteria (Mellor et al., 1994):

$$\frac{|H_{i+1} - H_i|}{(H_{i+1} + H_i)} \leq \alpha, \tag{1}$$

where $H_{i+1}$ and $H_i$ are the depths at two adjacent grids, and $\alpha$ is a slope factor equal to 0.2.

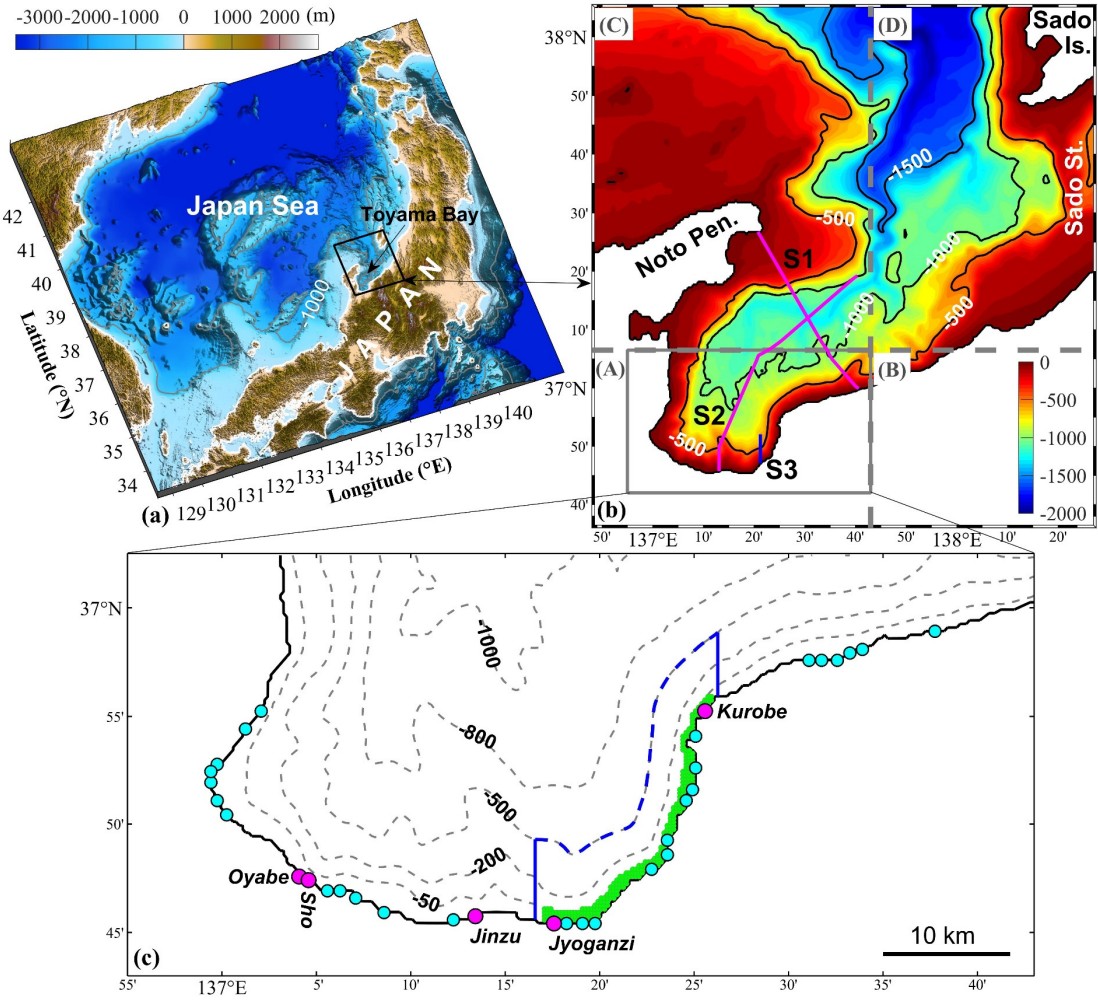

**Figure 1: (a)** Study area. **(b)** Model domain for Toyama Bay. Colors and contour lines represent water depth, with unit in meters. The model domain was divided into four areas from (A) to (D) with different horizontal resolutions. The respective horizontal resolutions are (1) 300 m × 300 m, (2) 900 m × 300 m, (3) 300 m × 900 m, and (4) 900 m × 900 m. "Pen.", "Is.", and "St." is the abbreviation of Peninsula, Island, and Strait respectively. The three lines S1, S2, and S3 represent the sections to show vertical distributions of the results. **(c)** The inner part of the bay. The circles indicate river mouths that supply freshwater and nutrients, with pink for first-class rivers and blue for second-class rivers. The green area means the location of submarine groundwater discharge (SGD). A uniform distribution of SGD over the grid points in this green area is assumed. The area inside the blue lines is used to calculate the inventories and material flows of SGD- and river-derived nutrients.

The model was driven by forces at the surface boundary and lateral boundary. The surface forcing was calculated based on the hourly output of the grid point value of the mesoscale model (GPV-MSM) provided by the Japan Meteorological Agency (Grid Point Value of Meso-Scale Model) with a resolution of 1/16° in zonal direction and 1/20° in meridional direction. The surface wind stress was calculated based on the difference of 10-m height wind velocity from GPV-MSM and the surface current velocity from our hydrodynamic model with the drag coefficient given by Mellor and Blumberg (2004). The surface heat flux consists of four terms: shortwave radiation, longwave radiation, sensible heat flux, and latent heat flux. The surface shortwave radiation was calculated utilizing the cloud cover data from the GPV-MSM and the formula in Rosati and Miyakoda (1988). The downward penetration of the shortwave radiation was calculated by the analytical formula given by Paulson and Simpson (1977). The longwave radiation, latent heat flux, and sensible heat flux were calculated by the bulk formula given by Kondo (1975) using the 1.5-m air temperature, 1.5-m relative humidity, cloud cover, 10-m wind speed data from the GPV-MSM, and the sea surface temperature from our hydrodynamic model. In addition, the precipitation from the GPV-MSM was specified in the model.

The lateral boundary conditions at the Japan Sea side, including surface elevation, current velocity, temperature, and salinity, were obtained by the bilinear interpolation of daily model output from the Japan Coastal Ocean Predictability Experiment (JCOPE-T) system developed by the Japan Agency for Marine-Earth Science and Technology (JAMSTEC) with a resolution

of 1/36º in both zonal and meridional directions (Varlamov et al., 2010). The lateral boundary conditions at the land side are the river discharges and SGD. In Japan, rivers are classified into two main categories based on their significance, scale, and management structure: first class rivers, which are important for national land conservation and the economy and are managed by the national government, and second class rivers, which are significant for regional public interests and are managed by prefectural governments. We specified the discharges of five first-class and 28 second-class rivers (Table S1), whose daily values were obtained from a terrestrial hydrology model developed by the River Engineering and Hydrology Laboratory, Chuo University (Matsuura and Tebakari, 2022). This model reproduced both the seasonal variations and the short-term fluctuations (Figure 2a). The mean value of the total river discharge into the bay is 295.4 $m^3 s^{-1}$, in which the first-class rivers accounted for approximately 78% (Figure 2a).

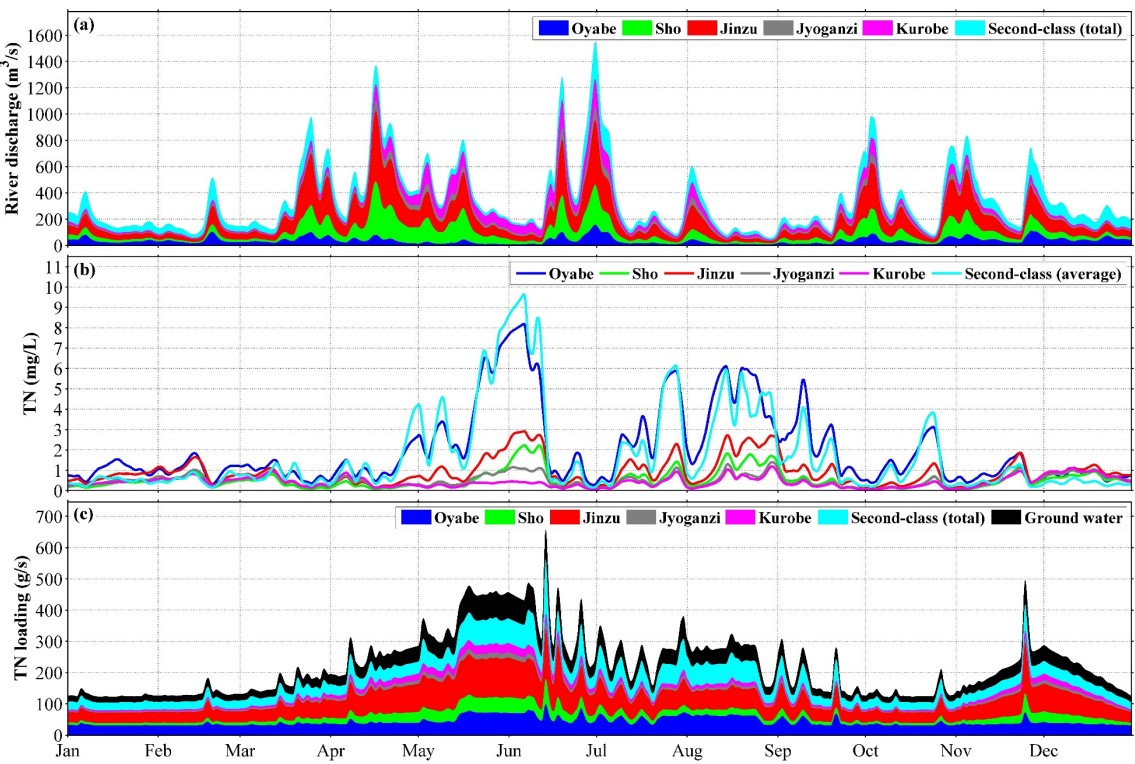

**Figure 2: (a) River discharge in 2016, (b) Total nitrogen (TN) concentration in the rivers, (c) TN loading from rivers. Dissolved inorganic nitrogen (DIN) concentration was specified to be 70% of TN concentration.**

Given that SGD predominantly occurs in the eastern coastal area of the bay (Hatta and Zhang, 2013), we set the outlet location of SGD at the sea bottom in an area between the Joganzi and Kurobe Rivers and between the coastline and the 70-m isobath (Figure 1c). The SGD amount into the bay is estimated to be 15% of the total discharge of nearby rivers, including the Joganzi and Kurobe Rivers, and 10 second-class rivers (Hatta et al., 2005; Hatta and Zhang, 2013) and has the same temporal variation as the river discharges. Consequently, we obtained a mean SGD of 12.6 $m^3 s^{-1}$. The corresponding flow velocity of SGD is about 5.7 $cm\ day^{-1}$, which is consistent with previous observations (Zhang et al., 2005). Because this value is small, we did not consider its mass contribution to the bay. However, we included its buoyancy effect by extracting a bottom flux of salinity with a formula $RS_b$, where R is the amount of SGD and $S_b$ is the bottom salinity. As we will show later, introducing the SGD's buoyancy effect is a key to the upward movement of the SGD-derived nutrients. Additionally, we did not used a moving land boundary in the model due to the minimal tidal range in the Japan Sea.

The model was run for two years (2015-2016) from the rest state with initial temperature and salinity fields interpolated from the JCOPE-T. We treated 2015 as a spin-up and used the results of 2016 for our analysis.

## 2.2 Ecosystem model

To simulate seasonal variations in nutrients, phytoplankton, zooplankton, and detritus (NPZD), we used an NPZD-type model, which has been used to simulate seasonal variations in nutrient and primary production in the Japan Sea (Onitsuka et al., 2007; Onitsuka and Yanagi, 2005). This nitrogen-based model has a similar structure to that of Fasham et al. (1990). The reason for choosing this ecosystem model is provided in Text S2. It comprises five compartments, namely, dissolved inorganic nitrogen (DIN), dissolved inorganic phosphate (DIP), phytoplankton (PHY), zooplankton (ZOO), and detritus (DET). The five biological compartments obey the same advection and diffusion equations as the water temperature but have an additional term representing biogeochemical processes, as follows:

$$\frac{\partial C}{\partial t} + Adv(C) = Diff(C) + Bio(C), \tag{2}$$

$$Adv(C) = u\frac{\partial C}{\partial x} + v\frac{\partial C}{\partial y} + w\frac{\partial C}{\partial z}, \tag{3}$$

$$Diff(C) = \frac{\partial}{\partial x}\left(K_h \frac{\partial C}{\partial x}\right) + \frac{\partial}{\partial y}\left(K_h \frac{\partial C}{\partial y}\right) + \frac{\partial}{\partial z}\left(K_v \frac{\partial C}{\partial z}\right), \tag{4}$$

where $C$ is the concentration of a state variable, $t$ is time, $Adv(C)$ is the advection term in which $u, v, w$ are velocity components in three directions ($x, y, z$), $Diff(C)$ is the diffusion term, and $Bio(C)$ is the sum of all the biogeochemical processes, which follow the equations in Guo and Yanagi (1998). $K_h$ and $K_v$ represent the horizontal and vertical eddy diffusivities, respectively. A schematic diagram of the biogeochemical processes between these state variables is provided in Figure 3 and the equations for the biogeochemical processes are provided in Text S3. The parameters used in the ecosystem model were obtained from previous studies (Guo and Yanagi, 1998; Ishizu et al., 2019; Onitsuka et al., 2007; Onitsuka and Yanagi, 2005) and modified slightly as shown in Table S2.

The initial and open boundary conditions for the ecosystem model were from the monthly climatological results of a biogeochemical and carbon model (JCOPE_EC) for the Japan Sea and the North Western Pacific (Ishizu et al., 2019), which was based on the Japan Coastal Ocean Predictability Experiment 2 (JCOPE2) system with a horizontal resolution of 1/12° (Miyazawa et al., 2009).

The nutrient (DIN and DIP) loadings (Figures 2c and S1b for DIN and DIP, respectively) of the rivers were calculated using river discharge (Figure 2a) and nutrient concentration (Figures 2b and S1a) obtained from a terrestrial hydrology model (Matsuura et al., 2023). The ratio of nitrate to phosphate in all rivers has a mean value of 17.8, which is a little larger than the Redfield ratio (N: P=16:1) (Redfield et al., 1963). The DIN loading of the first-class rivers accounts for approximately 78% of the total riverine DIN loading, and the mean value of the total DIN loading is 133.4 $gs^{-1}$. The total DIN loading shows a clear seasonal variation, with high values in June and July and low values in January (Figure 2c). The SGD nutrient loading was considered to be the same as the sum of the nutrient loading of nearby rivers and kept the same N:P ratio (= the Joganzi and Kurobe Rivers and 10 second-class rivers) (Hatta et al., 2005) whose mean value of DIN was 26.7 $gs^{-1}$.

Based on estimation by previous studies (Itahashi et al., 2021), the atmospheric deposition flux of DIN in the study area is approximately 1.2 $gs^{-1}$. Since this nutrient input is much smaller than the riverine and SGD nutrient loads, we did not include it in the model. Additionally, some direct discharges (e.g., sewage and industrial discharges) are incorporated into the model as part of the riverine inputs (Matsuura et al., 2023). Seagrass and other benthic phytoplankton on the seabed are not included in the model, as they are not significant in the eastern part of the bay (Ministry of the Environment, 2008), where SGD occurs.

The coupled physical-ecosystem model was initiated on the first day of January 2015 and subsequently integrated for 2 years. The first year was treated as the spin-up of the ecosystem model, while the second year, when the model results showed a stable seasonal variation, was analyzed.

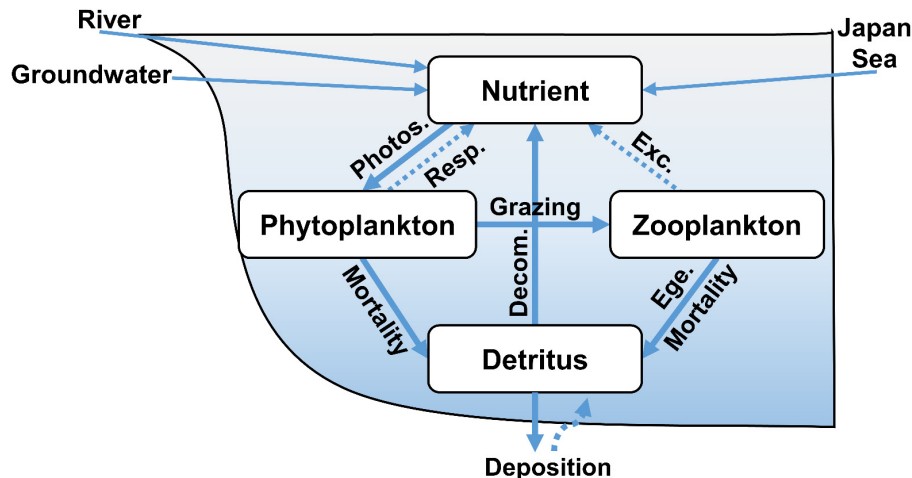

Figure 3. Biogeochemical processes in the ecosystem model. In this model, groundwater is released at the bottom of the sea. Photos., Resp., Exc., Ege., and Decom. represent Photosynthesis, Respiration, Excretion, Egestion, and Decomposition, respectively.

## 2.3 Tracking module

To evaluate the respective contributions of external nutrients to total nutrient and primary production in the bay, a tracking module was embedded into the coupled physical-ecosystem model (Kawamiya, 2001; Ménesguen et al., 2006). In this tracking module, the state variables for different nutrient sources were calculated separately as was done in the original physical-ecosystem model. Because the primary production in the Japan Sea is primarily limited by nitrogen (Onitsuka and Yanagi, 2005), the tracking module was established for nitrogen.

In our study area, we considered three external nutrient sources: the Japan Sea (JS), rivers (RV), and SGD (GW). As an example for SGD, we used $DIN_{GW}$, $PHY_{GW}$, $ZOO_{GW}$, and $DET_{GW}$ to track the cycle of SGD-derived nutrients in the system. The difference between total nutrients to the sum of these three external nutrients was considered as residual nutrients (RE), which was also calculated because its disappearance means the perfect representation of JS, RV, and SGD for the nutrients in our study area.

In the calculation, each subset of variables was attributed to the same physical and biogeochemical processes as was the case for total nutrients. It has the same advection and diffusion terms as the total nutrients but with different boundary conditions that will be described later. The separation of biogeochemical terms was based on the ratio of one subset state variable to the sum of the same state variables in the original ecosystem model. For example, the term for one biogeochemical process from state variable A to state variable B, named $Bio^{A \to B}$ (A and B can be any of four state variables, i.e., DIN, PHY, ZOO, and DET), in the original ecosystem model, was separated using the ratio of state variable A in the process supported by a single nutrient source, $A_X$ (X represents any of JS, RV, GW, and RE) to the state variable A as a whole in the original ecosystem model, as described in Eq. (5).

$$Bio_X^{A \to B} = \frac{[A_X]}{[A]} \cdot Bio^{A \to B}, X = JS, RV, GW, RE \tag{5}$$

where $Bio_X^{A \to B}$ represents the biogeochemical process supported by a single source of nutrients. The brackets denote concentrations, with $[A]$ representing the sum of $[A_{JS}]$, $[A_{RV}]$, $[A_{GW}]$, and $[A_{RE}]$. The complete expression for each biogeochemical process supported by a single nutrient source in the tracking module is provided in Text S4.

The tracking module was integrated simultaneously with the coupled physical-ecosystem model for all the nutrients. The initial conditions for the concentration of the subset state variables corresponding to the three external nutrient sources were set to zero while those for the residual nutrients were the results of the coupled physical-ecosystem model for all the nutrients. For

the tracking calculation of the rivers and SGD-derived nutrients, we introduced their nutrient fluxes from the rivers and SGD, respectively; both processes have no nutrient fluxes from the Japan Sea at the open boundaries. For the tracking calculation of the nutrients from the Japan Sea, it has the only supply from the open boundaries; for the tracking calculation of the residual nutrients, it has zero supply from rivers, SDG, and the open boundaries. The other configurations for the tracking module were the same as those for the original ecosystem model. The integrations for each tracking process also lasted for two years to obtain a stable seasonal variation state.

## 3 Results

The results of the physical-ecosystem model were validated by the observational data, including temperature, salinity, DIN, and Chlorophyll-a concentration from the Marine Information Research Centre (MIRC) Ocean Dataset 2005, provided by the MIRC, Japan Hydrographic Association (MIRC, 2005). The original data were collected from 1934 to 2001, and the spatial-temporal coverage of data was higher for temperature and salinity than for DIN and Chlorophyll-a. Due to the limitations in concurrent observational data availability, direct validation becomes challenging. We processed the original data to the standard layer depth and averaged the data spatially following the model grid and temporally following the month. The gridded monthly climatological data were then compared with the model results. The positions of the observational data are plotted along with the surface distributions of model results (Figure 4), and the quantity of data available for each point is shown in Figure S2.

The correlation coefficients and root mean square deviations (RMSD) between the observational data and the model results in different months are listed in Table S3. The mean correlation coefficients (RMSD) of temperature, salinity, DIN, and Chlorophyll-a in all months are 0.92 (1.75 °C), 0.64 (0.26), 0.93 (2.25 mmolN m$^{-3}$), and 0.65 (0.13 mg m$^{-3}$), respectively. The comparisons for different depths are shown in the scatter plots (Figures S3a–S3d). There are some discrepancies between model results and field observations, which likely result from their different time periods and that the model did not account for temporal changes in environmental and anthropogenic factors. Based on previous observational records, the long-term temperature increase in the western Japan Sea over the past 100 years has been estimated at approximately +1.51°C (Japan Meteorological Agency, 2024). Additionally, while the population around Toyama Bay has increased over the decades (Aoki, 2021), the implementation of new wastewater treatment systems has resulted in a 50% reduction in riverine nutrient inputs despite relatively stable river discharge (Katazakai and Zhang, 2021a). Furthermore, climate change-induced shifts from snowfall to rainfall have altered the SGD recharge patterns and chemical composition, leading to lower nutrient concentrations in SGD (Katazakai and Zhang, 2021b). Both the long-term warming trend and the reduction in nutrient inputs contribute to the discrepancies between our model results and the historical observational data. Additionally, the low temporal resolution of some observational data, with only a few values recorded per month, may also contributed to these discrepancies (Skogen et al., 2021). However, the spatial patterns and seasonal variations of the physical and biological fields shown in the model results were in good agreement with the observational data.

### 3.1 Physical fields

The eastward coastal branch of the Tsushima Warm Current (TWC) flows into the upper layer of Toyama Bay, and the inflow shows two patterns within the year. From April to August, it enters the bay along the west coast, resulting in higher salinity in the western part of the bay, whereas in other months, it tends to flow southeastward across the bay mouth (Figures 4 and S4). These two inflow patterns are consistent with those reported by previous studies (Igeta et al., 2017, 2021; Nakada et al., 2005). Additionally, the surface flow pattern in the inner part of the bay is counterclockwise along the coast from west to east (Figure S4).

Affected by the inflow of freshwater from the land side, the surface salinity in the inner part of the bay is always low, and the low salinity surface water spreads out along the eastern coast (Figure 4). In April, the freshwater discharge increases with the melting of snow (Figure 2a), which increases the area of low-salinity surface water in the bay. Then, with the entry of the high-salinity coastal branch of the TWC along the western coast, the low-salinity surface water is concentrated in the eastern part of the bay. With the second increase in freshwater discharge in July, the area of the low-salinity water expands again (Figure 4). In the following months toward the winter, the low-salinity surface water begins to mix downward, reaching about 200 m in January (Figures 5 and S5). The low-salinity water can spread out to the outer part of the bay and even distribute near the Sado Strait (Figure 1) along the eastern coast (Figures 4 and 5). The intrusion of the high-salinity water from the TWC in the intermediate layer from the subsurface to about 200 m begins in May and continues until December (Figure 5).

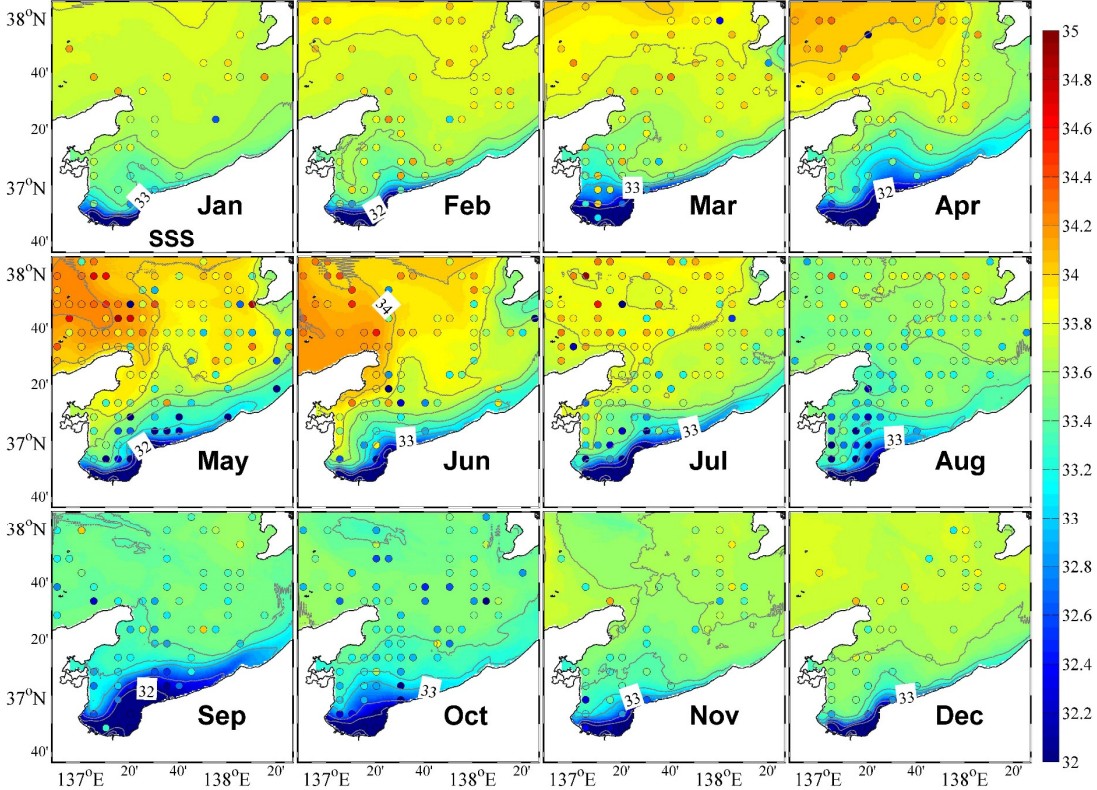

Figure 4: Monthly mean surface salinity. The circles represent observational data, whose color bar is shown on the right.

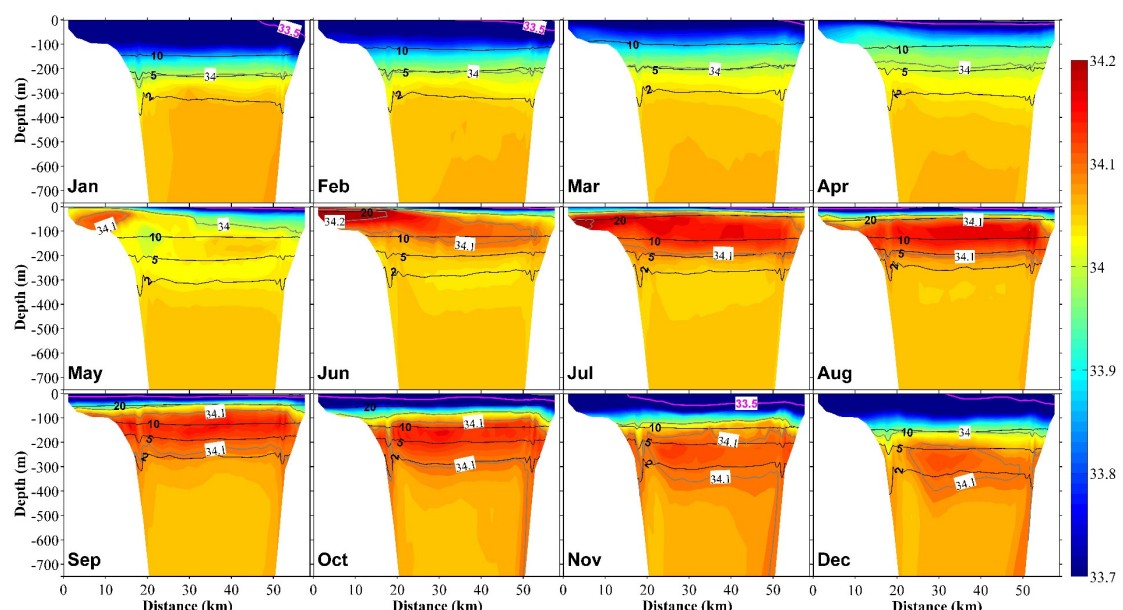

**Figure 5: Vertical distribution at section S1 (Figure 1b) of salinity (color) and temperature (black contours) in 12 months. The direction of the spatial axis represents the transect along the S1 line from north to south, as shown in Figure 1b, with the distance measured from the northern end of the S1 line. The pink contour line represents a salinity of 33.5.**

The water temperature in the upper layer (0-200 m) shows remarkable seasonal variations, with stratification in summer (starting from May) and mixing in winter (starting from October). Below the middle layer (>200 m), a permanent thermocline exists (Figure 5). The water in the bay consists of three different water masses, namely, the coastal surface layer (< 50 m), with low salinity resulting from the freshwater discharge into the bay from rivers; the TWC water mass (at a depth of ~200 m) with high salinity originating from the Tsushima Current; and the Japan Sea Proper water mass, which is cold, highly saline, and dominates the deep layers of the Japan Sea (Hatta et al., 2005).

### 3.2 Biological fields

Our model showed that the surface distribution of DIN was similar to that of the low-salinity surface water (Figure 6), and the vertical distribution of DIN in the central part of the bay showed open-ocean characteristics, with low and high nutrient concentrations in the surface and lower layers, respectively. Further, we also consistently obtained high nutrient concentrations along the inner coastal area of the bay throughout the year (Figures 6 and S6). From winter to spring, the upward supply of nutrients from the lower layer owing to vertical mixing promoted the increase of nutrients in the surface layer (Figure S6), which is a necessary condition for the appearance of phytoplankton spring bloom.

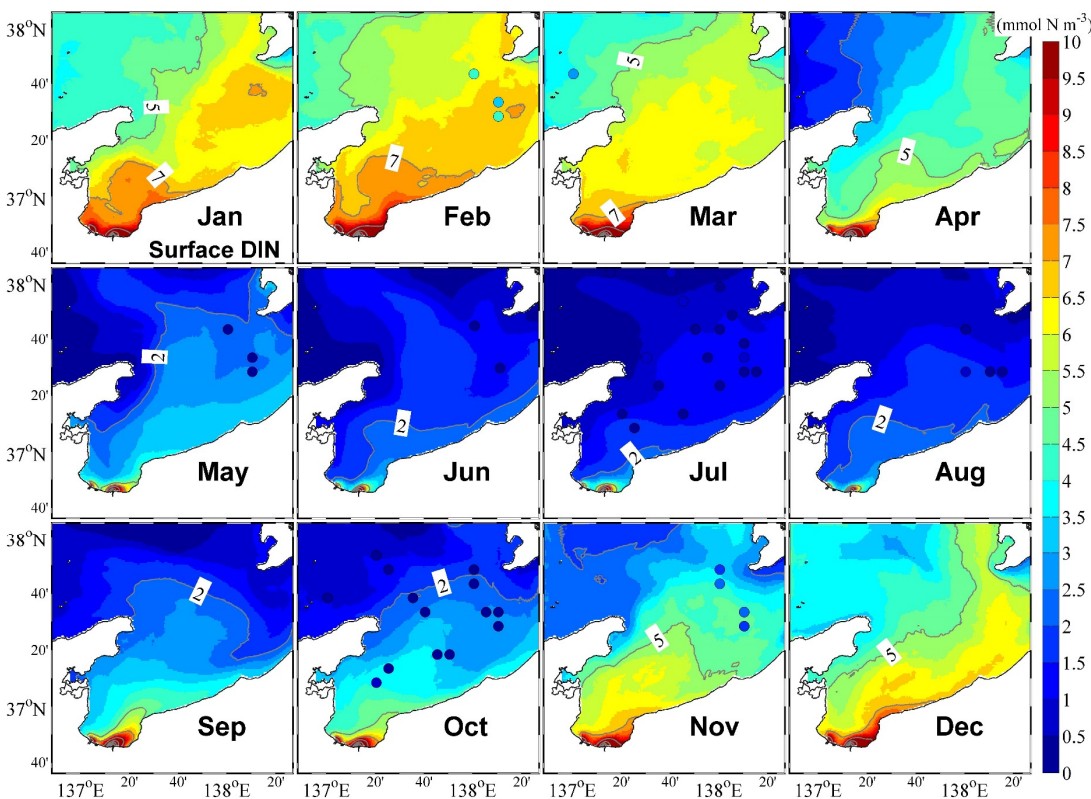

**Figure 6: Monthly mean surface dissolved inorganic nitrogen (DIN). The color circles represent observational data.**

Our observations also indicated that in the central bay area, phytoplankton are primarily distributed in the upper 100-m layer, with the upper 50-m layer characterized by a remarkable spring bloom with a high abundance of phytoplankton (Figures 7 and 8). From May, surface phytoplankton abundance decreased throughout the bay, except in the estuary area where nutrients are supplied from the landside (Figure 7). Further, in May, the maximum subsurface chlorophyll concentration, which persisted till September, was observed at a depth of approximately 30 m (Figure 8). The autumn bloom occurred in September and was slightly weaker than the spring bloom (Figures 7 and 8). From late autumn to winter, the growth of phytoplankton is restricted

due to light and temperature limitations, causing a decrease in the concentration of phytoplankton in the whole area (Figures 7 and 8). The decrease in phytoplankton during this period promotes nutrients from the rivers to accumulate in the estuary area (Figures 6 and S6).

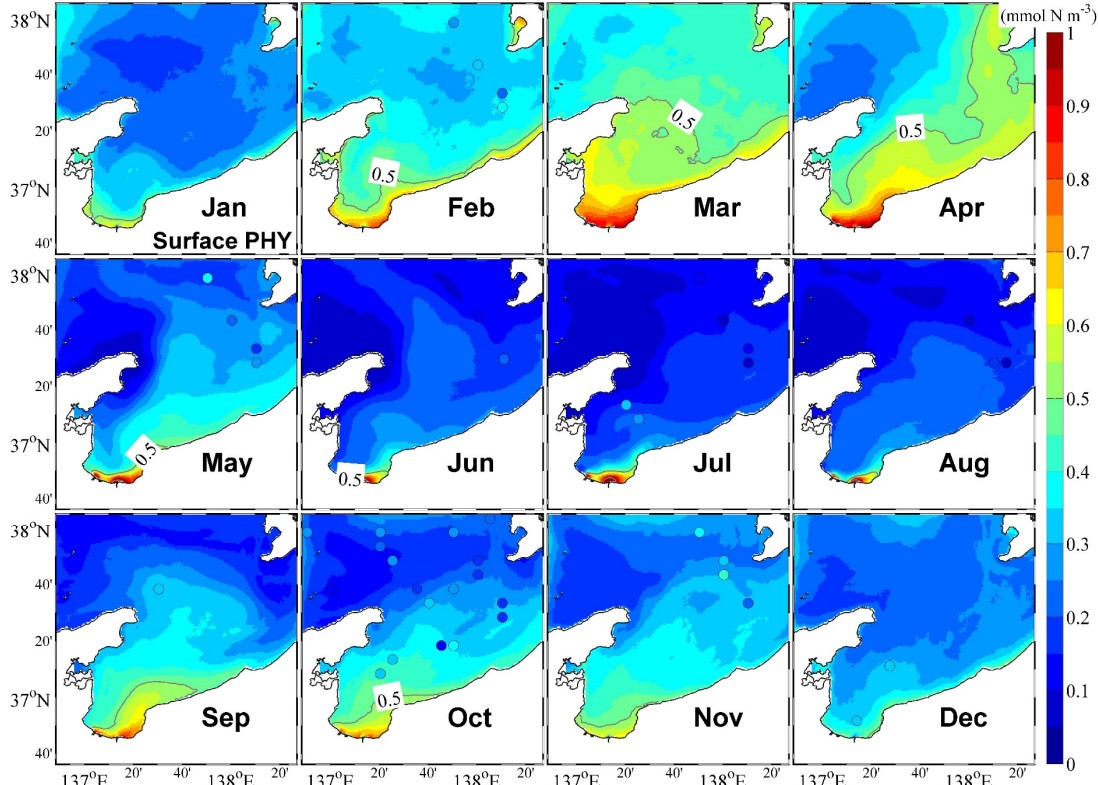

**Figure 7: Monthly mean surface phytoplankton concentrations. The color circles represent observational data.**

The distributions and seasonal variations of zooplankton are strongly dependent on the phytoplankton (Figures S7). Similarly, the changes in detritus (derived from phytoplankton and zooplankton) coincide with the changes in phytoplankton, except that the concentrations of surface detritus are lower in shallower areas near the shore, possibly due to the sinking of detritus (Figures S8).

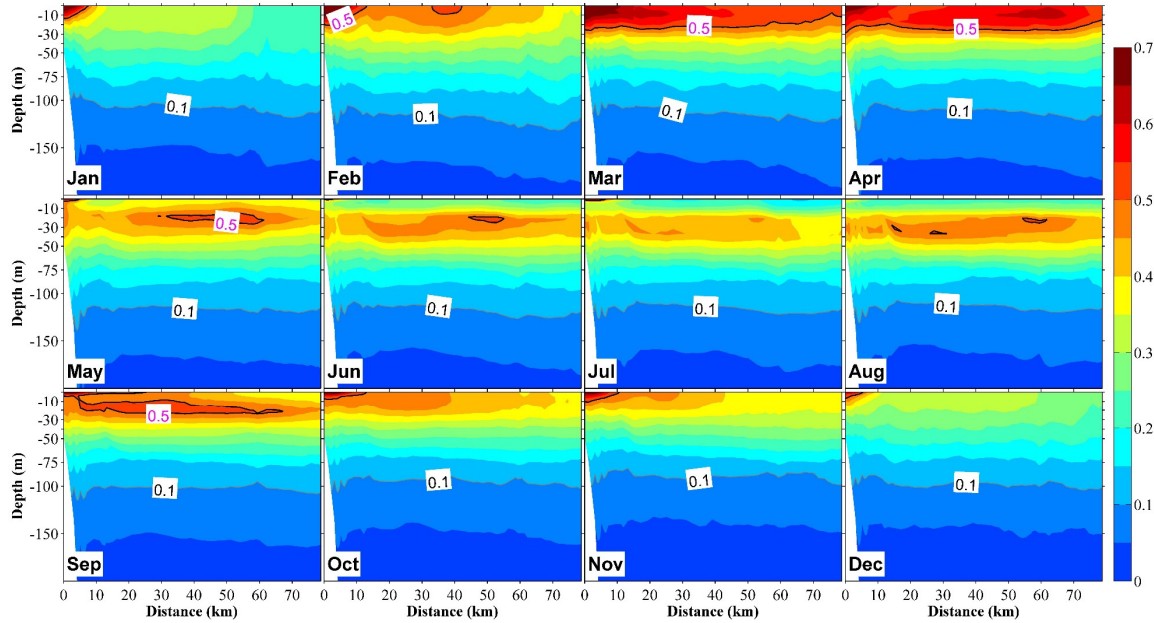

**Figure 8: Vertical distribution at section S2 (in Figure 1b) of phytoplankton in 12 months, with the unit in mmolN $m^{-3}$. The distance on the x-axis is measured from the shore.**

## 3.3 Contributions of different nutrient sources to total nutrient concentrations and primary production

Based on model results given by the tracking module, which enabled the distinction of nutrients originating from different sources, we evaluated the respective contributions of different nutrient sources to total nutrients and primary production. We found that nutrients derived from river water were predominantly distributed in the surface layer (0–20 m) of the inner coastal area of the bay. These river-derived nutrients spread out to the central part of the bay, reaching ∼40 km from the shore (Figures 9 and S9). Further, these river-derived nutrients accounted for more than 50% of total nutrients close to the river mouths (∼20 km from the shore) (Figure 9), and the seasonal variation of this proportion (Figures 9 and S9) was consistent with that of riverine nutrient loading (Figure 2c), which peaked in June and July.

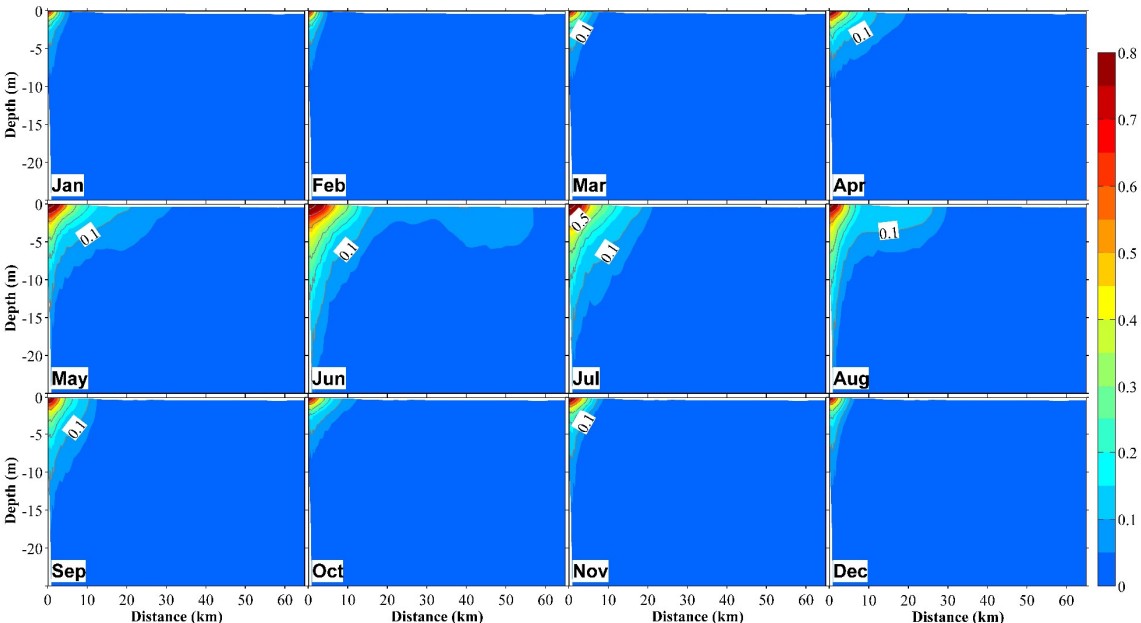

**Figure 9: Vertical distribution at section S2 (Figure 1b) of the ratio of dissolved inorganic nitrogen originating from river water (DIN$_{RV}$) to the total DIN in 12 months. The distance on the x-axis is measured from the shore.**

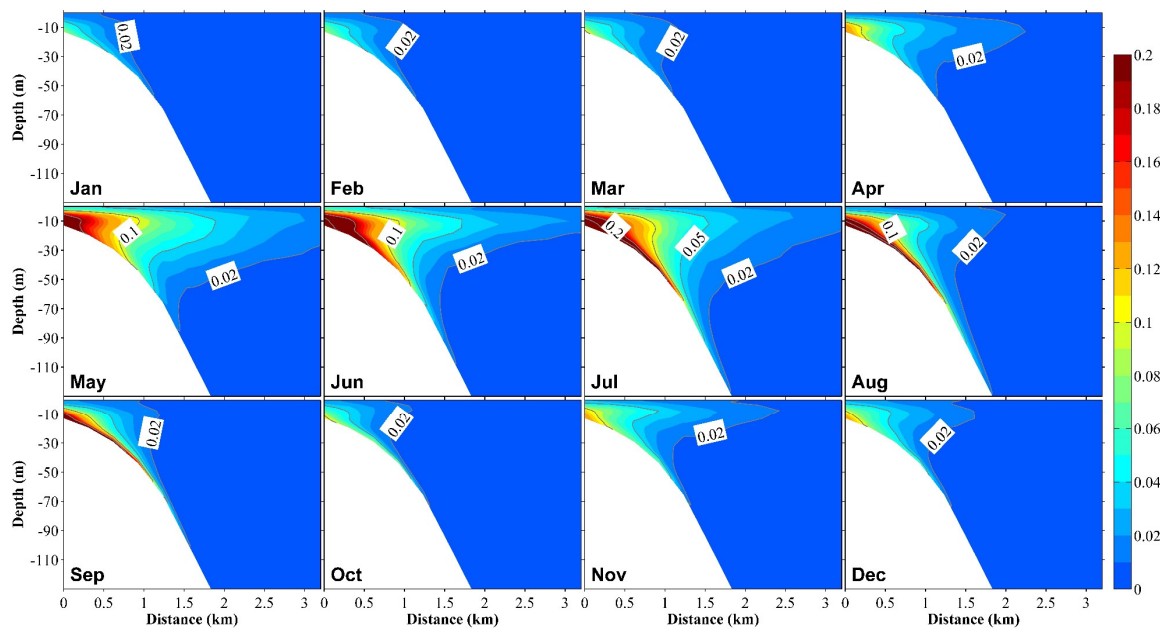

**Figure 10: Vertical distributions at section S3 (Figure 1b) of the ratio of dissolved inorganic nitrogen originating from submarine groundwater discharge ((DIN$_{SGD}$) to the total DIN in 12 months. The distance on the x-axis is measured from the shore.**

On the other hand, SGD-derived nutrients were primarily distributed in areas close to the coast within ~3 km from the shore, and they accounted for a smaller proportion of the total nutrients relative to river-derived nutrients; their maximum contribution to total nutrients close to the coast reached ~20% (Figures 10 and S10). Additionally, SGD-derived nutrients were not only abundant in the middle and bottom layers (>5 m depth) but also showed upward movement to the surface layer throughout the year (Figure 10). Like river-derived nutrients, SGD-derived nutrients showed their highest contribution to total nutrients in June and July (Figure 10). Except for the inner coastal areas, the nutrients from the Japan Sea dominated in the other areas (Figure S11).

Given that nutrients derived from the landside (including river water and SGD) were found to be predominantly distributed in the inner coastal area, we divided the Toyama Bay area (Figure 1c) into two sub-areas, the shallow and deep water sub-areas, based on the 100-m isobath line, which is ~4 km away from the shore. Next, we calculated the volume-averaged DIN in each sub-area from a depth of 70 m to the surface for each nutrient source (Figure 11). In the shallow water sub-area, the annual-volume-averaged concentration of total DIN was 5.59 mmolN m$^{-3}$, with contributions from river water, SGD, and the Japan Sea accounting for 0.81 mmolN m$^{-3}$ (14.5%), 0.25 mmolN m$^{-3}$ (4.5%), and 4.53 mmolN m$^{-3}$ (81%), respectively. In the deep water sub-area, the volume-averaged concentrations of DIN derived from river water and SGD were very low, being 0.09 and 0.02 mmolN m$^{-3}$, respectively, while the DIN from the Japan Sea had a mean value of 5.05 mmolN m$^{-3}$. We also observed that the residual nutrients, which represent the initially present nutrients, in the bay were generally consumed within the first half year (Figure 11), implying a fast water exchange process of the Bay with the Japan Sea.

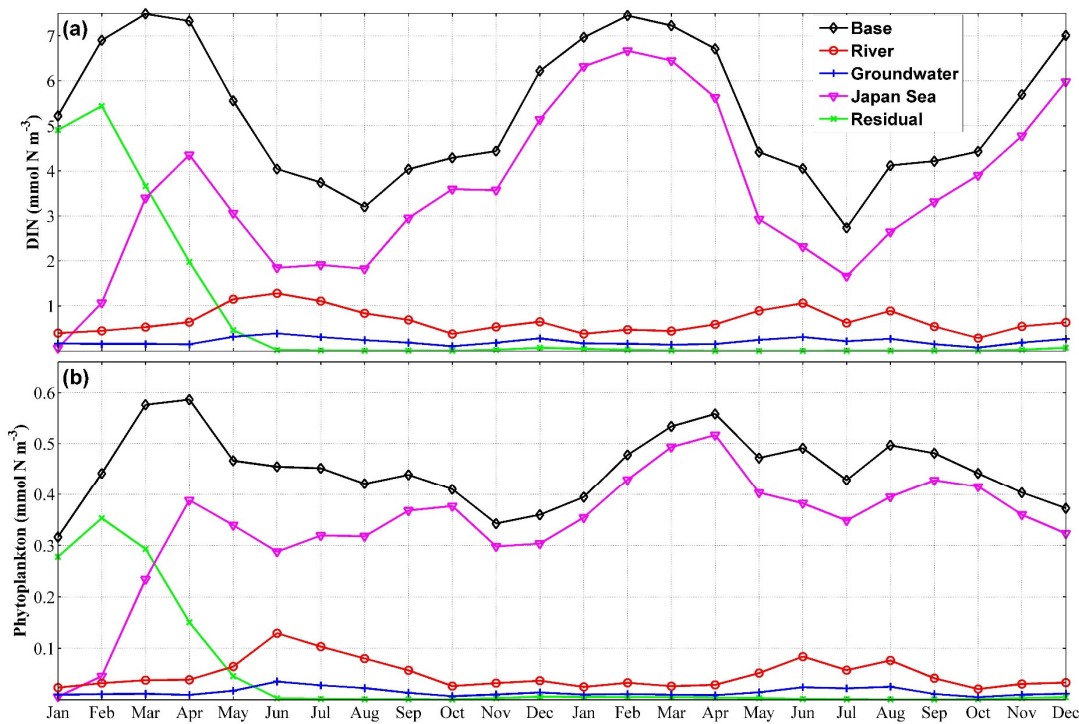

**Figure 11: Two-year time series of the volume-averaged concentrations of dissolved inorganic nitrogen (DIN) (a) and phytoplankton (b) in the shallow water sub-area from a depth of 70 m to the surface for each nutrient source. 'Base' represents the total concentration, while 'River', 'Groundwater', 'Japan Sea', and 'Residual' represent the nutrient source of river water, groundwater, Japan Sea, and the residual nutrients.**

The ratio of phytoplankton supported by river-derived nutrients to total phytoplankton was slightly smaller than the ratio of river-derived nutrients to total nutrients; however, their distributions and seasonal variations were similar (Figures 11, S12, and S13). The contributions of the SGD-derived nutrients to phytoplankton growth were also almost the same as their contributions to the total nutrients (Figures 12 and S14). From June to August, the contributions of river- and SGD-derived nutrients to phytoplankton growth are highest, exceeding 30% (within ~10 km off the shore) and 10% (within ~1 km off the

shore), respectively (Figures 12 and S13). In the shallow water sub-area, the annual-volume-averaged concentration of total phytoplankton was 0.456 mmolN m$^{-3}$, with phytoplankton supported by nutrients from river water, SGD, and the Japan Sea contributing 0.055 mmolN m$^{-3}$ (12%), 0.018 mmolN m$^{-3}$ (4%), and 0.383 mmolN m$^{-3}$ (84%), respectively (Figure 11). Furthermore, in the deep water sub-area, the volume-averaged concentrations of phytoplankton supported by landside-derived nutrients, including river water and SGD, decreased to a negligible value.

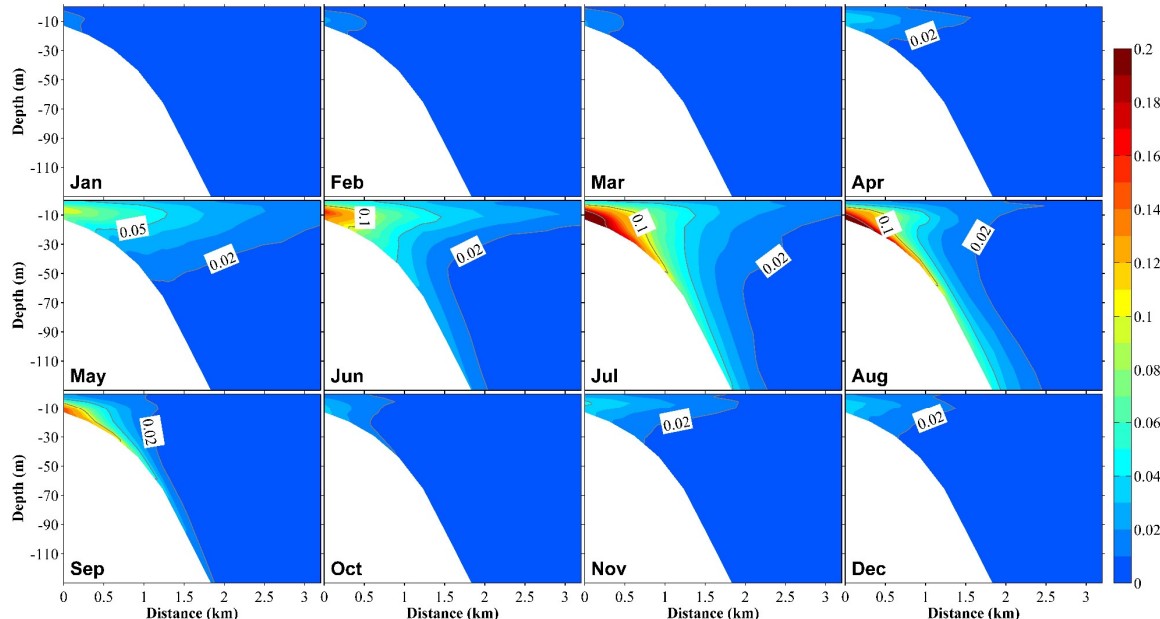

**Figure 12: Vertical distributions at section S3 (Figure 1b) of the proportions of phytoplankton supported by nutrients originating from submarine groundwater discharge (PHY$_{SGD}$) to the total phytoplankton in 12 months. The distance on the x-axis is measured from the shore.**

## 4 Discussion

### 4.1 Impact of the buoyancy effect of SGD on upward nutrient transport

As stated in Section 3.3, although SGD-derived DIN was released from the sea bottom, it can reach the surface layer. One possible mechanism responsible for this process is the buoyancy effect of SGD. The fresh SGD is lighter than surrounding seawater at the sea bottom and causes upward convection, which promotes the upward transport of nutrients from the sea bottom (Kreuzburg et al., 2023). To confirm its effect, we performed another simulation (Case 2), in which we removed the bottom salinity flux due to SGD in the hydrodynamic model but kept the SGD-derived DIN flux in the ecosystem model. For easy reference, we call the calculation with the bottom salinity flux due to SGD as Case 1.

The upward vertical current was weaker in Case 2 than in Case 1 (Figure S15). Since most previous studies only estimated the amount of SGD (Hatta and Zhang, 2013; Luoma et al., 2021; Santos et al., 2009, 2021; Xu et al., 2024b), we know little about its impact on the vertical current in the sea. The difference between the two calculations shows that the buoyance effect of SGD can intensify the vertical velocity by about 0.1~0.2 mm/s (Figure S15). In addition, the salinity above the outlet location of SGD slightly decreased by 0.1 – 0.4 with the inclusion of bottom salinity flux due to SGD in Case 1 (Figure S15).

With the weakening of the upward vertical current, the upward transport of SGD-derived nutrients was also weakened in Case 2 with respect to Case 1. Figure 13 shows the vertical distributions of the annual mean nutrients originating from SGD and their supported phytoplankton of Case 1 and Case 2, and the differences in the nutrients and phytoplankton between these two calculations. Without the buoyancy of SGD, the SGD-derived nutrients were mainly distributed from the sea bottom to the subsurface layer, and their distribution was almost absent in the surface layer (Figure 13). In Case 2, the SGD-derived nutrients were supplied to the surface layer only in winter. In addition, without the buoyancy effect of SGD, more nutrients spread away

from the shallow water area in the bottom layer (Figure 13). Similarly, the phytoplankton supported by SGD-derived nutrients
decreased in Case 2 (Figure 13).

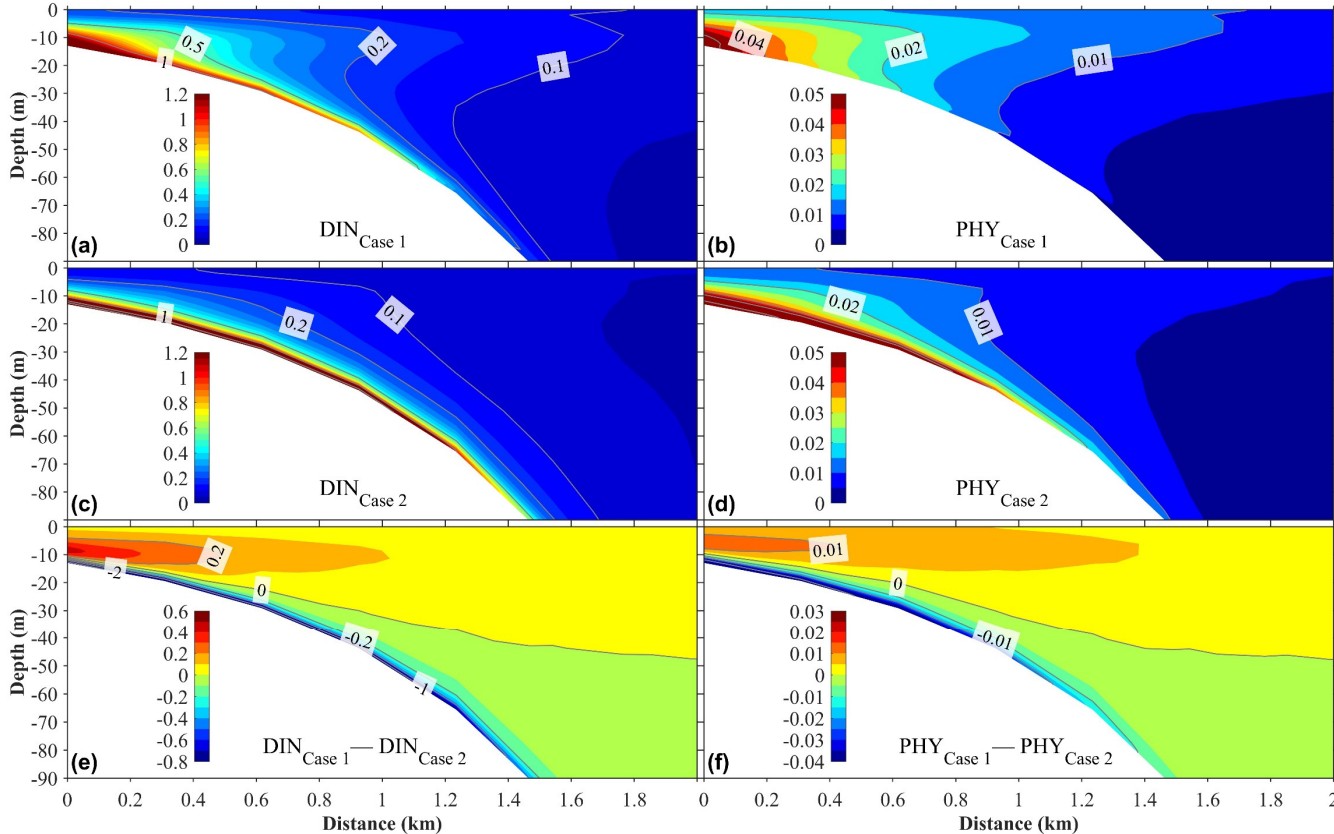

**Figure 13: (a)−(d) illustrate the vertical distributions of the annual mean dissolved inorganic nitrogen (DIN$_{SGD}$) originating from submarine groundwater discharge (SGD) and the phytoplankton (PHY$_{SGD}$) it supports under two scenarios. Panels (a) and (b) correspond to Case 1, where the buoyancy effect of SGD is included in the hydrodynamic model, while panels (c) and (d) correspond to Case 2, where the buoyancy effect of SGD is excluded. (e) and (f) show the differences in DIN$_{SGD}$ and PHY$_{SGD}$ between Case 1 and Case 2. The distance on the x-axis is measured from the shore.**

Our additional calculation shows that in a deep coastal sea, the buoyancy effect of SGD plays an important role in the upward transport of SGD-derived nutrients from the sea bottom. Without the buoyancy effect of SGD, it is difficult for these nutrients to reach the euphotic layer and be consumed by phytoplankton. Therefore, in addition to the wind-induced and tide-induced vertical mixing, the buoyancy effect of SGD is also an important mechanism for the upward transport of SGD-derived nutrients.

**4.2 Comparison of utilization between SGD-derived and riverine nutrients**

Weather and how much SGD-derived nutrients can be used by phytoplankton should depend on the water depth because the concentration of SGD-derived nutrients generally decreases from the sea bottom to the sea surface. To evaluate the utilization of SGD-derived nutrients by the phytoplankton, we calculated the related material flows in four areas near the SGD outlet locations (Figure 1c) with different water depths, i.e., 0−20 (area 1), 20−70 (area 2), 70−200 (area 3), and 200−500 m (area 4) (Figure 14). The net biogeochemical consumption flux of DIN is defined as,

$$Net\ biogeochemical\ processes\ of\ DIN =$$

$$-uptake + respiration + mineralization + excretion. \quad (6)$$

The annual mean transport of SGD-derived DIN into the bay is 1925 mmol s$^{-1}$. Since the 0−20 m area occupies part of the SGD outlet, the DIN transport into this area is 433 mmol s$^{-1}$. Among the biogeochemical processes, the consumption rates (photosynthesis) of SGD-derived DIN in the 4 areas from shallow to deep were 94, 120, 43, and 70 mmol s$^{-1}$, and the regeneration rates including respiration, excretion, and mineralization were 32, 65, 83, and 119 mmol s$^{-1}$, respectively (Figure

14). After excluding the vertical export flux of detritus (DET) to the sediment shown in Figure 14, the horizontal transport of biological particles (PHY+ZOO+DET) related to SGD-derived DIN is 48 mmol s[-1] from area 1 to area 2, 98 mmol s[-1] from area 2 to area 3, 57 mmol s[-1] from area 3 to area 4, and 8 mmol s[-1] from area 4 to the further offshore area. Such a reduction in horizontal transport of the particles with the distance away from the coast is consistent with the reduction of utilization of SGD-derived DIN from area 1 to area 4. Apparently, with the increasing of water depth where SGD-derived DIN distributes from area 1 to area 4, it becomes difficult for SGD-derived nutrients to reach the euphotic layer and be consumed by phytoplankton (Figure 14b).

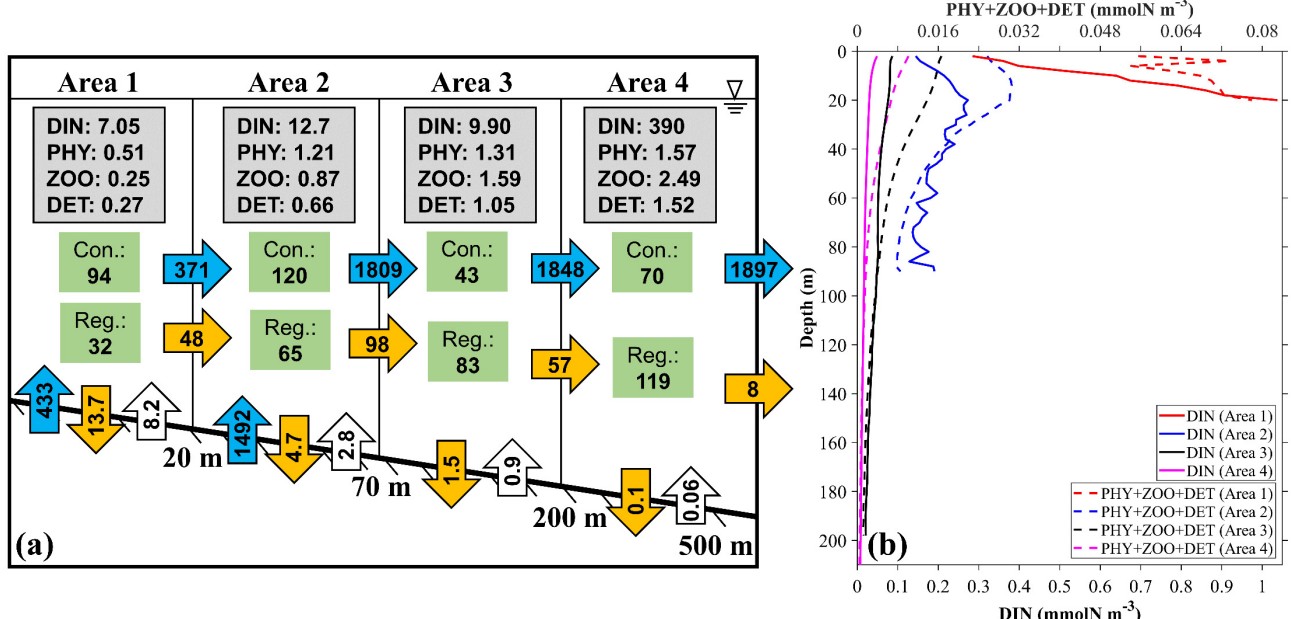

Figure 14: (a) The annual mean inventories and material flows of SGD-derived nutrients in 4 different areas near the outlet locations of SGD (Figure 1c). (Area 1) 0−20 m, (Area 2) 20−70 m, (Area 3) 70−200 m, (Area 4) 200−500 m. The values in the grey rectangles represent the inventory (×10[7] mmol) of the ecosystem variables in the area. The values in the green rectangles represent the consumption (Con.) and regeneration (Reg.) rates of $DIN_{GW}$ in the biogeochemical processes (mmol s[-1]). The values in the vertical blue arrows represent the input transport of $DIN_{GW}$ (mmol s[-1]). The values in the vertical orange and white arrows represent the vertical flux to the sediment and re-decomposition flux from the sediment to the sea (mmol s[-1]). The values in the horizontal blue and orange arrows represent the horizontal transport of $DIN_{GW}$ and its related biological particles ($PHY_{GW}$+$ZOO_{GW}$+$DET_{GW}$) (mmol s[-1]). (b) show the average profile of $DIN_{GW}$, $PHY_{GW}$, and $DET_{GW}$ in these 4 areas.

Similarly, we also calculated the material flow of river-derived nutrients in the 4 areas (Figure 15). Among the biogeochemical processes, the consumption rates of river-derived DIN in the 4 areas were 286, 235, 238, and 542 mmol s[-1] and the regeneration rates were 125, 168, 218, and 445 mmol s[-1], respectively (Figure 15), all of which are larger than those related to SGD-derived DIN by several times. The horizontal transport of the biological particles (PHY+ZOO+DET) supported by river-derived DIN was 114 mmol s[-1] from area 1 to area 2, 176 mmol s[-1] from area 2 to area 3, 194 mmol s[-1] from area 3 to area 4, and 291 mmol s[-1] from area 4 to the further offshore area. The biological activity related to river-derived DIN was kept and even increased from area 1 to area 4, and consequently, the proportion of riverine nutrients used by phytoplankton increased. Differing from the SGD-derived DIN that stayed in the deep layer, the river-derived DIN was mainly distributed in the surface layer (Figure 15b) and therefore easily consumed by phytoplankton.

Currently, the impact of landside-derived nutrients (including SGD and rivers) on coastal marine ecosystems is usually evaluated by their nutrient loadings, without considering their respective distributions in the sea and whether they can be really used by the phytoplankton (Santos et al., 2021; Silverman et al., 2024). Our results show that due to the different distribution characteristics of nutrients originating from different sources (SGD and rivers), their contribution rates to the phytoplankton growth also vary with water depth. Additionally, in addition to the fresh SGD we studied here, there is another type of SGD in the seas that comes from saline SGD, i.e. recirculating seawater (Santos et al., 2021; Xu et al., 2024b). The saline SGD are widely distributed in the seabed deeper than fresh SGD and release mostly recycled nutrients to global coastal waters (Santos

et al., 2021). However, due to the lack of buoyancy in saline SGD and the deep location where it enters the sea, the nutrients derived from saline SGD would be more difficult than the fresh SGD to be transported upward and then consumed by phytoplankton. Therefore, when evaluating the impact of SGD-derived nutrients on coastal marine ecosystems, it is necessary to consider not only the loading but also the distribution of SGD in which the fresh and saline SGDs are different.

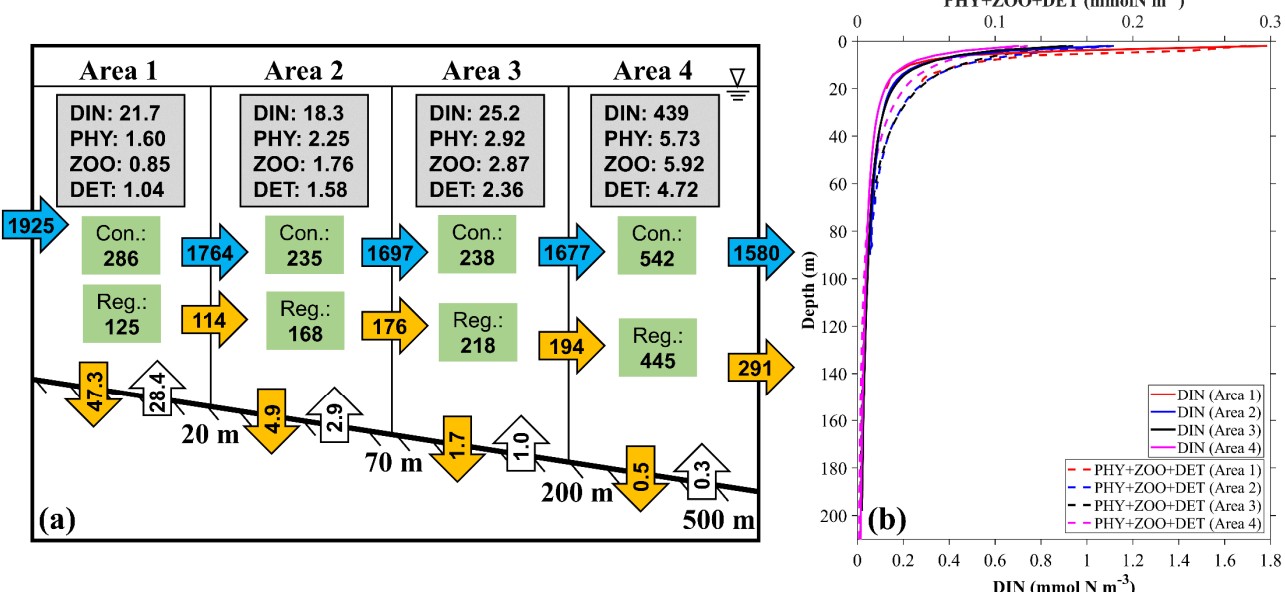

**Figure 15: (a) The annual mean inventories and material flows of river-derived nutrients in 4 different areas near the outlet locations of SGD (Figure 1c). (Area 1) 0−20 m, (Area 2) 20−70 m, (Area 3) 70−200 m, (Area 4) 200−500 m. The values in the grey rectangles represent the inventory (×$10^7$ mmol) of the ecosystem variables in the area. The values in the green rectangles represent the consumption (Con.) and regeneration (Reg.) rates of $DIN_{RV}$ in the biogeochemical processes (mmol s$^{-1}$). The values in the far left horizontal blue arrow represents the input transport of $DIN_{RV}$ (mmol s$^{-1}$). The values in the vertical orange and white arrows represent the vertical flux to the sediment and re-decomposition flux from the sediment to the sea (mmol s$^{-1}$). The values in the horizontal blue and orange arrows represent the horizontal transport of $DIN_{RV}$ and its related biological particles ($PHY_{RV}$+$ZOO_{RV}$+$DET_{RV}$) (mmol s$^{-1}$). (b) show the average profile of $DIN_{RV}$, $PHY_{RV}$, and $DET_{RV}$ in these 4 areas.**

In this study, we evaluated the spatial distribution of SGD-derived nutrients and their contribution to phytoplankton growth in Toyama Bay, which experiences a high rate of groundwater discharge. Consequently, in other regions with similar conditions, the presented results can serve as an upper limit of the effect of SGD-derived nutrients in a coastal bay area. However, we did not consider benthic phytoplankton or seagrass on the seabed, and the distribution of SGD varies with changes in water depth. Given the complexity of environmental conditions in different marine regions where benthic phytoplankton or macrophytes are important, or where SGD is located at shallower depths, the contribution of nutrients derived from SGD to phytoplankton growth could be higher. Indeed, in some regions, SGD-derived nutrients can contribute to eutrophication (Luijendijk et al., 2020). For example, in shallow coastal bays such as Liaodong Bay (Luo et al., 2023) and Zhenzhu Bay (Xu et al., 2024a), where the average water depth is less than 30 meters, SGD-derived nutrients may exert a more significant impact and can also lead to eutrophication. Additionally, in areas where seagrass meadows or benthic microalgae are present, SGD can strongly influence the biotic characteristics of seagrass beds (Kantún-Manzano et al., 2018).

## 5 Conclusions

In this study, we constructed a coupled physical-ecosystem model with a tracking module to evaluate the effects of submarine groundwater discharge (SGD) derived nutrients as well as nutrients from other external sources on total nutrients and phytoplankton growth in Toyama Bay, where the SGD flow rates are greater than are observed in most other areas worldwide (Taniguchi et al., 2002; Zhang et al., 2005). The distributions and spatiotemporal characteristics of the impacts of these nutrients from different sources were clarified using the tracking results. Additionally, we clarified the seasonal variations in

the total nutrient dynamics of Toyama Bay as well as its ecosystem structure and characteristics. High concentrations of nutrients are distributed along the coastal area of the bay throughout the year. Phytoplankton are primarily found in the upper 100 m, with two blooms occurring in spring and autumn. From May to September, a subsurface chlorophyll maximum appears at approximately 30 m.

Specifically, our model revealed that in this bay, nutrients derived from river water are predominantly distributed in the surface layer (0–20 m) close to river mouth areas (~20 km from the shore), with an annual average contribution of 14.5% to the total nutrient inventory within a narrow band from the coastline. Conversely, nutrients derived from SGD were primarily distributed in areas close to the coast (~3 km from the shore), contributing 4.5% to the total nutrient inventory within a narrow band from the coastline, which is lower than the contribution from river-derived nutrients. It was also found that SGD-derived nutrients released from the sea bottom were abundant in the middle and bottom layers (>5m depth) and some of them could move upward to the surface layer. Comparison based on simulations with and without the buoyancy effect of SGD in the hydrodynamic model revealed that the upward transport of SGD-derived nutrients to the surface layer is primarily attributed to the buoyancy effect of SGD. The spatial distribution range of SGD-derived nutrients became smaller in the calculation results without the buoyancy effect of SGD, implying that the buoyancy effect of SGD cannot be ignored when evaluating the behavior of SGD-related nutrients.

The distributions and seasonal variations of phytoplankton supported by river- and SGD-derived nutrients were similar to the characteristics of the respective nutrient sources. From June to August, the contributions of river- and SGD-derived nutrients to phytoplankton growth are highest, exceeding 30% and 10%, respectively. The annual average phytoplankton biomass supported by river- and SGD-derived nutrients contributed 12% and 4%, respectively, to the total phytoplankton biomass within a narrow band from the coastline. The different distribution depths of river- and SGD-derived nutrients in the water column determine their contribution to the phytoplankton growth from the coastal to the offshore areas. The river-derived nutrients were surface-orientated and easily used by the phytoplankton in a larger area from the coast. In contrast, the SGD-derived nutrients were bottom-orientated and were not easily used by the phytoplankton. In the area close to the coast, the shallow water depth allows it to fall within photic zone, making it accessible for use by phytoplankton. With increasing distance from the coast, the nutrients' limited dispersal to offshore euphotic zones gradually makes them less available for phytoplankton use.

An important implication of this study is that the evaluation of the role of SGD-derived nutrients in the marine ecosystem cannot be based on the input amount of nutrients. A more reasonable way is based on how much nutrients can be used by the phytoplankton and enter the biogeochemical cycle of nutrients in the sea. This does not deny the importance of landside-derived nutrients to the marine ecosystem because the area affected by landside-derived nutrients usually is the spawning area of many marine organisms such as firefly squids, krills, and halfbeaks in the bay (Iguchi, 1995; Oya et al., 2002). It also provides a reference for nutrient management.

**Author contribution**

MD developed the coupled physical-ecosystem model and performed the simulations. MD and XG prepared the manuscript. All the authors reviewed the manuscript.

**Competing interests**

The contact author has declared that none of the authors has any competing interests.

## Acknowledgments

This work was supported by the Environment Research and Technology Development Fund (JPMEERF20212001) of the Environmental Restoration and Conservation Agency of Japan and a Grant-in-Aid for Scientific Research (MEXT KAKENHI, grant number: 22H05206). M. Dong was supported by the Ministry of Education, Culture, Sports, Science and Technology, Japan (MEXT) to a project on Joint Usage/Research Center–Leading Academia in Marine and Environment Pollution Research (LaMer).

## Data Availability Statement

The model results that support the findings of this study are available online at https://doi.org/10.5281/zenodo.11075610.

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
