# Peer review of "Effects of Submarine Groundwater on Nutrient Concentration and Primary Production in a Deep Bay of the Japan Sea"

_EGUsphere, 2024_

## Referee Comment (RC2)

**Review** of the manuscript *Effects of submarine groundwater on nutrient concentration and primary production in a deep bay of the Japan Sea*, by M. Dong, X. Guo, T. Matsuura, T. Tebakari and J. Zhang, submitted to **Biogeosciences**

**Manuscript overview**

Th manuscript present a modelling study into the origin of marine nutrients in a semi-enclosed bay off the west coast of Japan. It tracks nutrients from both riverine sources and fresh ground water sources and quantifies their importance to the local nutrient budget and the local primary production.

**Review overview**

The manuscript is generally well written though the objectives and main results could be more clearly presented. The presented method is not new (as clearly stated), but is applied to an area where is has not been used before and to a marine nutrient input source (fresh ground water) that has not been considered before. As such I find that the presented work merits publication. The manuscript is accompanied by an extensive appendix which I have not considered. The presented results are very local and more effort could be made to derive conclusions for other areas. At the same time the authors could make their local results more clear (percentage contributions, area affected) in the abstract and conclusions. Validation of simulated results with observations from the same period would be desirable, as would be discussion on other nutrient input sources (atmospheric deposition, direct discharges, aquaculture). More detailed comments are provided below.

**Recommendation**

Minor revision

**Detailed Comments**

1. Line 18: "*narrow band of the coastline (< 3km)*", this results is also mentioned in the conclusions but with less detail. As the (short) abstract should be a reflection of the main findings of the presented work, and the (longer) conclusions likewise, I would expect the abstract to have the same or less information as the conclusions section, not more.

2. Line 18: "*middle and bottom layers*", please specify these layers as this is very location specific.

3. Line 55-57: this technique was also applied in Northwest Atlantic areas, see Lenhart & Große (2018), OSPAR (2013) and Painting et al (2013). Please acknowledge this work as well.

4. Line 60: and hence eutrophication issues.

5. Line 61: I would prefer to see a more elaborate site description here. What does the seabed consist of mainly? What is the general circulation pattern, is there inflow of deep water from the Japan sea? Just to get a physical feel for the area.

6. Line 65: "*are greater is this area than in most areas worldwide*", that does make this site a good choice, but also means that the found percentage contribution from fresh ground water to primary production is likely a maximum compared to other sites known to have fresh ground water inputs. I miss this comparative observation in the manuscript.

7. Line 114: "*precipitation from the GVP-MSM was specified in the model*", was atmospheric deposition of nutrients also considered in the model? If not, please state so clearly.

8. Line 119: here the land-based inputs are specified as being the riverine contributions and the fresh ground water contributions. But what about other sources? The bay seems to be densely populated (Toyama city alone has a population of over 400,000) so I would expect significant direct discharges in the area (e.g. sewage discharges, industrial discharges). These are generally lower in nutrient loads than rivers but could be of the same order of magnitude as the ground water loads. Naturally these discharges would be at the surface, and thus mixed with the riverine signal, but some information on these sources would be beneficial. Is there any aquaculture activity in the bay that could add to the nutrient loads?

9. Line 120: please specify what constitutes a first class and a second class river.

10. Line 154: for me the model schematic belongs in the main text, not the supplementary materials. And does the biogeochemical model have a name?

11. Line 162: "*The nutrient (DIN and DIP) loadings (Figure 2c)*", Figure 2c only shows DIN, not DIP. I would prefer to see DIP also.

12. Table 1: the information is appreciated but for me this could be in the supplementary materials.

13. Line 184-186: am I correct in assuming that any initially present nutrients (pelagic, benthic) represent the residual term, and that these are replaced by the named nutrient sources over time as the residual goes to zero? Does this mean there is no long term storage of nutrients in the sea bed?

14. Line 211: why are the model results from 2015 and 2016 validated using observations gathered from 1934 to 2001? Surely more recent observations are available? Given the rise in global temperature across the 20th century which is continuing to accelerate in the 21st century I would expect a discord between historic observations and current simulations. Not to mention the population increase in the area over the observational time span and since. I think the authors are doing themselves and their model a disservice like this. It may be that more recent observations are not available, but this should be discussed in the main text: discrepancies between the model results (2016) and the observations (1935-2001) are not necessarily the models fault.

15. Line 222: it would help to have a visual overview of the main (horizontal) current patterns in the bay.

16. Line 234: month → months?

17. Figure 3: it would be good to know the temporal resolution of the observations being compared to the simulated monthly average. Some text on this should be included in the main text. A simulated monthly average is unlikely to validate well against observations if these amounted to a handful a month. See also Skogen et al (2021).

18. Figure 4: please indicate the direction of the spatial axis. Distance from where? This figure could also be enlarged to show more detail in the euphotic zone.

19. Line 245: which are stratified in summer → with stratification in summer.

20. Line 271: yes, not surprising in an NPZD model. Is there any dynamic mortality for zooplankton included that could cause differences between zooplankton dynamics and that of their food source?

21. Figure 7: shows an autumn bloom that is not mentioned in the text anywhere. And the transect starts at the mouth of the Jinzu river, correct? Again, it is not mentioned what the distance from refers to.

22. Line 283: as transect S2 seems to start at a river mouth it is not surprising that riverine nutrients account for more than 50% of nutrients close to shore. I would have expected it to be more. Is the rest input from the Japan Sea?

23. Line 315: please provide the annual average percentages. ? And would it be possible to add a spatial plot with the average, annual, depth-integrated contributions from rivers and SGD to primary production in addition to figure 10?

24. Discussion: in general I miss a discussion here about the transferability of these results to other sites. What about sites which have benthic phytoplankton or macrophytes on the seabed? Or larger rivers which reach more offshore areas? The limitations of the applied model (no benthic phytoplankton, no nutrient storage in the sea bed it seems) should be discussed in this light. An NPZD model is after all a relatively simple model. This also applies to the statement at the beginning which said this site experiences high ground water discharges. Does that mean the found values (percentages contribution from rivers and SGD) can be seen as maximums with regard to other sites?

25. Line 340: Case 2 than Case 1 → in Case 2 with respect to Case 1.

26. Line 365: shouldn't the term "*photosynthesis*" be replaced by "uptake"?

27. Figure 12: the standard figure sub-numbering is left to right, then down, not down and then right. Figure 12e should read Case 2.

28. Figure 14: great figure, and I would prefer to see the equivalent graph for riverine sources in the main text.

29. Line 408: please make the conclusions section self-explanatory by avoiding acronyms like SGD without explanation.

30. Line 412-413: this result mentioned in the conclusions is only supported by evidence in the supplementary materials. I would argue that any conclusion presented here must be supported by material included in the main text.

31. Line 415: please define "close to river mouth areas"

32. Line 417: please specify the contribution of SGD nutrients in percentage to the total, this is the main objective of the manuscript.

33. Line 419: I would say it was based on simulations with and without the buoyancy effect, as it is in essence the same model.

34. Line 426: "*the shallow water depth allows for inclusion in the photic zone and thus use by phytoplankton* "

35. Line 427: please rephrase to make it clear you are still talking about the SDG-derived nutrients, what distance from the coast you are referring to and why it would be difficult for phytoplankton to use them. Surely you are referring to the lack of dispersal of these nutrients to offshore euphotic areas? That does not make it difficult for plankton to use them, it simply means they have no access to them. Given the objective of this manuscript, listing the relative contributions here (in %) and the area affected (in km) by them should be a priority.

**References**

Lenhart, H. J., & Große, F. (2018). Assessing the effects of WFD nutrient reductions within an OSPAR frame using trans-boundary nutrient modeling. *Frontiers in Marine Science*, *5*, 447.

OSPAR (2013) Distance to target modelling assessment, report 2013-599, ISBN 978-1-909159-32-7, https://www.ospar.org/documents?v=7319

Painting, S. J., Van der Molen, J., Parker, E. R., Coughlan, C., Birchenough, S., Bolam, S., Aldridge, J.N., Forster, R.M. & Greenwood, N. (2013). Development of indicators of ecosystem functioning in a temperate shelf sea: a combined fieldwork and modelling approach. *Biogeochemistry*, *113*(1), 237-257.

Skogen, M.D., Ji, R., Akimova, A., Daewel, U., Hansen, C., Hjollo, S.S., van Leeuwen, S.M., Maar, M., Macias, D., Mousing, E.A., Almroth-Rosell, E., Sailley, S.F., Spence, M.A., Troost, T., van de Wolfshaar, K. (2021) *Disclosing the truth: are models better than observations?*, Marine Ecology Progress Series, DOI: 10.3354/meps13574

---

## Author Comment (AC1)

*Reply to Referee #2:*

*Manuscript overview*

*Th manuscript present a modelling study into the origin of marine nutrients in a semi-enclosed bay off the west coast of Japan. It tracks nutrients from both riverine sources and fresh ground water sources and quantifies their importance to the local nutrient budget and the local primary production.*

*Review overview*

*The manuscript is generally well written though the objectives and main results could be more clearly presented. The presented method is not new (as clearly stated), but is applied to an area where is has not been used before and to a marine nutrient input source (fresh ground water) that has not been considered before. As such I find that the presented work merits publication. The manuscript is accompanied by an extensive appendix which I have not considered. The presented results are very local and more effort could be made to derive conclusions for other areas. At the same time the authors could make their local results more clear (percentage contributions, area affected) in the abstract and conclusions. Validation of simulated results with observations from the same period would be desirable, as would be discussion on other nutrient input sources (atmospheric deposition, direct discharges, aquaculture). More detailed comments are provided below.*

    Thanks very much for your helpful comments and suggestions. In this study, we did not consider benthic phytoplankton, such as seagrass, on the seabed, and the distribution of SGD varies with changes in water depth. Given the complexity of environmental conditions in different marine regions where benthic phytoplankton or macrophytes are important, or where SGD is located at shallower depths, the contribution of nutrients derived from SGD to phytoplankton growth may be higher. We have strengthened the comparison of our results with those from other regions in the revised manuscript. Additionally, we also enhanced the presentation of our findings in the abstract and conclusions.

    Regarding the validation of simulated results using observations from the same period. Thank you for your suggestion. However, due to the limited availability of observational data from the same period, it is challenging to perform a direct comparison. Regarding nutrient inputs in Toyama Bay, the main sources are rivers and groundwater from the landward side, and inputs from the Japan Sea on the ocean side. Contributions from other sources, such as atmospheric is minimal. Therefore, we did not include these in our model. We have addressed these issues in the revised manuscript.

    Our responses to the more detailed comments are as follows. The referee's comments are cited in italics.

*Recommendation*

*Minor revision*

Thank you for the positive evaluation.

***Detailed Comments***

*Line 18: "narrow band of the coastline (< 3km)", this results is also mentioned in the conclusions but with less detail. As the (short) abstract should be a reflection of the main findings of the presented work, and the (longer) conclusions likewise, I would expect the abstract to have the same or less information as the conclusions section, not more.*

Thank you for pointing this out. We have strengthened the description of the main findings in the conclusions section and modified the abstract in the revised manuscript.

*Line 18: "middle and bottom layers", please specify these layers as this is very location specific.*

Thank you for your comment. These layers refer to the areas located approximately 5 m below the sea surface. We also added this clarification in the revised manuscript.

*Line 55-57: this technique was also applied in Northwest Atlantic areas, see Lenhart & Große (2018), OSPAR (2013) and Painting et al (2013). Please acknowledge this work as well.*

Thank you for your suggestion. We have cited these papers in the revised manuscript to acknowledge their contributions.

*Line 60: and hence eutrophication issues.*

Thanks. We have added this presentation in the revised manuscript.

*Line 61: I would prefer to see a more elaborate site description here. What does the seabed consist of mainly? What is the general circulation pattern, is there inflow of deep water from the Japan sea? Just to get a physical feel for the area.*

The following two websites provide detailed information about the geology and deep water characteristics of Toyama Bay. We also provided an introduction to the physical field in Section 3.1 based on the model results.

https://gbank.gsj.jp/geonavi/geonavi.php#11,36.93757,137.25200

https://t-deepsea.jp/en/deepsea/about/

Toyama Bay stretches approximately 46 km from east to west and 74 km from south to north. It has an average water depth of about 550 meters, with a maximum depth reaching 1,114 meters. The seabed primarily consists of soft sediments, including mud and silt, particularly in the deeper regions. Shallower areas have sandy or gravelly substrates, often influenced by riverine inputs from nearby rivers such as the Jinzu, Sho, and Kurobe Rivers. The water in the bay is made of three different water

masses: the coastal surface layer water (~50m) with low salinity due to the freshwater from the river and precipitation, the Tsushima Warm Current (TWC) water (~ 200m) with high salinity, and the deep water (Japan Sea Proper water). The surface circulation in the inner part of Toyama Bay follows a counterclockwise pattern along the coast, moving from west to east. This circulation is influenced by the eastward coastal branch of the TWC, which flows into the upper layer of the bay. The deep water primarily remains separated from the surface waters by the TWC water and does not mix easily with the shallower layers.

*Line 65: "are greater is this area than in most areas worldwide", that does make this site a good choice, but also means that the found percentage contribution from fresh ground water to primary production is likely a maximum compared to other sites known to have fresh ground water inputs. I miss this comparative observation in the manuscript.*

Thank you for your comments. The reported SGD flow rates in the eastern coastal area of the bay, observed as approximately $72–187$ $cm$ $day^{-1}$ (Zhang et al., 2005), are greater than the global average of $6.5$ $cm$ $day^{-1}$ (Santos et al., 2021). The dissolved inorganic nitrogen (DIN) and dissolved inorganic phosphorus (DIP) fluxes entering Toyama Bay via SGD, estimated using a box model, are $2.13$ $mmol$ $m^{-2}$ $per$ $day$ and $0.02$ $mmol$ $m^{-2}$ $per$ $day$, respectively (Hatta et al., 2005), which are comparable to the global averages of $4.06$ $mmol$ $m^{-2}$ $per$ $day$ and $0.06$ $mmol$ $m^{-2}$ $per$ $day$ (Santos et al., 2021). We have included a clarification of this comparative observation in the revised manuscript.

*Line 114: "precipitation from the GVP-MSM was specified in the model", was atmospheric deposition of nutrients also considered in the model? If not, please state so clearly.*

We did not consider atmospheric nutrient deposition in the model. Thank you for pointing this out, and we have made this clear in the revised manuscript.

*Line 119: here the land-based inputs are specified as being the riverine contributions and the fresh ground water contributions. But what about other sources? The bay seems to be densely populated (Toyama city alone has a population of over 400,000) so I would expect significant direct discharges in the area (e.g. sewage discharges, industrial discharges). These are generally lower in nutrient loads than rivers but could be of the same order of magnitude as the ground water loads. Naturally these discharges would be at the surface, and thus mixed with the riverine signal, but some information on these sources would be beneficial. Is there any aquaculture activity in the bay that could add to the nutrient loads?*

Thank you for your comments. Our riverine nutrient loads have accounted for some nutrient contributions from industrial and sewage discharges. We provided the necessary explanations regarding these sources in the revised manuscript. In addition, there is some aquaculture activity in Toyama Bay, but its scale and impact on nutrient loads appear limited. The region's nutrient dynamics are more significantly influenced by inputs from rivers and SGD, which contribute considerable amounts of nutrients from land to the bay.

*Line 120: please specify what constitutes a first class and a second class river.*

Thank you for highlighting this. Regarding these rivers, we included a table (Table S1) in the revised supplementary materials to list all the river names along with their discharge and nutrient loads.

*Line 154: for me the model schematic belongs in the main text, not the supplementary materials. And does the biogeochemical model have a name?*

Thank you for your feedback. We agree that the model schematic is an essential part of the explanation and should be included in the main text rather than the supplementary materials. We relocated it accordingly in the revised manuscript. Regarding the biogeochemical model, it does not have a specific name but is a customized NPZD-type model developed for this study.

*Line 162: "The nutrient (DIN and DIP) loadings (Figure 2c)", Figure 2c only shows DIN, not DIP. I would prefer to see DIP also.*

We provided a figure (Figure S1) showing the DIP loadings in the revised supplementary materials.

*Table 1: the information is appreciated but for me this could be in the supplementary materials.*

Thank you for your input. We agreed with your suggestion and moved this table to the revised supplementary materials as Table S2.

*Line 184-186: am I correct in assuming that any initially present nutrients (pelagic, benthic) represent the residual term, and that these are replaced by the named nutrient sources over time as the residual goes to zero? Does this mean there is no long term storage of nutrients in the sea bed?*

Thank you for your observation. Yes, the initially present nutrients can be considered as the residual term, which diminishes over time as they are replaced by the named nutrient sources. Long-term storage of nutrients in the sea bed was not considered in our model, as its contribution is minimal.

*Line 211: why are the model results from 2015 and 2016 validated using observations gathered from 1934 to 2001? Surely more recent observations are available? Given the rise in global temperature across the 20th century which is continuing to accelerate in the 21st century I would expect a discord between historic observations and current simulations. Not to mention the population increase in the area over the observational time span and since. I think the authors are doing themselves and their model a disservice like this. It may be that more recent observations are not available, but this should be discussed in the main text: discrepancies between the model results (2016) and the observations (1935-2001) are not necessarily the models fault.*

Thank you for raising this point. We agree with you that using observational data from a different time period may introduce discrepancies, given the potential changes in climate, human activities, and environmental conditions over time. Unfortunately, the availability of concurrent observational data is limited, which makes direct validation challenging.

To address this, we opted to use multi-year historical data as the best available reference for validation. We have now included a discussion in the revised manuscript to explain this limitation and to clarify that discrepancies between the model results and the historical observations are not necessarily due to model inaccuracies but may also reflect temporal changes in environmental and anthropogenic factors.

*Line 222: it would help to have a visual overview of the main (horizontal) current patterns in the bay.*

Thank you for your suggestion. The current field is shown in Figure S4 of the revised supplementary materials.

*Line 234: month → months?*

Thanks. We have made this change in the revised manuscript.

*Figure 3: it would be good to know the temporal resolution of the observations being compared to the simulated monthly average. Some text on this should be included in the main text. A simulated monthly average is unlikely to validate well against observations if these amounted to a handful a month. See also Skogen et al (2021).*

Thank you for your suggestion and the reference. Since the number of observations varies across locations, we have included a distribution figure (Figure S2) in the revised supplementary materials that shows the quantity of observation data available for each point.

*Figure 4: please indicate the direction of the spatial axis. Distance from where? This figure could also be enlarged to show more detail in the euphotic zone.*

Thank you for pointing this out. The direction of the spatial axis represents the transect along the S1 line from north to south, as shown in Figure 1b, with the distance from the northern end of the S1 line. In the revised manuscript, we have added this clarification to the figure caption. We have also enlarged the figure to show more detail in the euphotic zone.

*Line 245: which are stratified in summer → with stratification in summer.*

Thanks. We have made this change in the revised manuscript.

*Line 271: yes, not surprising in an NPZD model. Is there any dynamic mortality for zooplankton included that could cause differences between zooplankton dynamics and that of their food source?*

Thank you for your comments. In this study, the focus has been primarily on nutrient and phytoplankton dynamics, so we have not incorporated dynamic mortality for zooplankton. As a result, zooplankton dynamics closely follow the availability of their food sources (such as phytoplankton). However, incorporating dynamic mortality could provide more realism by introducing factors such as predation, starvation, and environmental stress, which might decouple zooplankton dynamics from their food sources. We would like to treat this modeling effort as our future works.

*Figure 7: shows an autumn bloom that is not mentioned in the text anywhere. And the transect starts at the mouth of the Jinzu river, correct? Again, it is not mentioned what the distance from refers to.*

Thank you for pointing this out. Regarding the autumn bloom, we updated the text to mention this event in the revised manuscript.

As for the transect, it is correct that it starts from the mouth of the Jinzu River. We have noted your suggestion and clarified the distance reference in the figure caption to make it more precise.

*Line 283: as transect S2 seems to start at a river mouth it is not surprising that riverine nutrients account for more than 50% of nutrients close to shore. I would have expected it to be more. Is the rest input from the Japan Sea?*

Yes. The rest is primarily from the Japan Sea (Figure S11).

*Line 315: please provide the annual average percentages. ? And would it be possible to add a spatial plot with the average, annual, depth-integrated contributions from rivers and SGD to primary production in addition to figure 10?*

Thank you for your suggestion. The annual-volume-averaged concentration of total phytoplankton was 0.456 mmolN m$^{-3}$, with phytoplankton supported by nutrients from river water, SGD, and the Japan Sea contributing 0.055 mmolN m$^{-3}$ (12%), 0.018 mmolN m$^{-3}$ (4%), and 0.383 mmolN m$^{-3}$ (84%), respectively. We have provided these annual average percentages of nutrient contributions in the revised manuscript. Additionally, we included two spatial plots (Figure S12 and S14) that show the average, annual, depth-integrated contributions from rivers and submarine groundwater discharge (SGD) to primary production in the revised supplementary materials.

*Discussion: in general I miss a discussion here about the transferability of these results to other sites. What about sites which have benthic phytoplankton or macrophytes on the seabed? Or larger rivers which reach more offshore areas? The limitations of the applied model (no benthic phytoplankton, no nutrient storage in the sea bed it seems) should be discussed in this light. An NPZD model is after all a relatively simple model. This also applies to the statement at the beginning which said this site*

*experiences high ground water discharges. Does that mean the found values (percentages contribution from rivers and SGD) can be seen as maximums with regard to other sites?*

Thank you for pointing out this important aspect.

We agree that the discussion could benefit from addressing the transferability of these results to other locations with differing characteristics, such as sites with benthic phytoplankton, macrophytes, or larger rivers that extend their influence farther offshore. We have included a discussion (on lines 471-475 of the revised manuscript) of these factors, emphasizing the limitations of our applied NPZD model, which does not currently include benthic phytoplankton in the seabed. These limitations could influence how results are interpreted in different contexts.

Furthermore, geographical conditions can also have an influence on the utilization of nutrient sources originating from SGD. For example, in Toyama Bay, due to its substantial variations in water depth, SGD-derived nutrients in deeper areas are less likely to be utilized, especially in the absence of benthic phytoplankton. Consequently, it is difficult to consider these contributions as maximum values applicable to other locations.

Thank you for your valuable suggestion; we have incorporated this discussion in the revised manuscript.

*Line 340: Case 2 than Case 1 → in Case 2 with respect to Case 1.*

Thanks. We made this change in the revised manuscript.

*Line 365: shouldn't the term "photosynthesis" be replaced by "uptake"?*

Thank you for the suggestion. We have revised this word.

*Figure 12: the standard figure sub-numbering is left to right, then down, not down and then right. Figure 12e should read Case 2.*

Thank you for pointing this out. We have revised this figure following your comments.

*Figure 14: great figure, and I would prefer to see the equivalent graph for riverine sources in the main text.*

Thank you for the suggestion. We have moved the equivalent graph for riverine sources to the revised manuscript, now labeled as Figure 15.

*Line 408: please make the conclusions section self-explanatory by avoiding acronyms like SGD without explanation.*

Thank you for the comments. We have revised the conclusions section to ensure it is self-explanatory and clear for all readers.

*Line 412-413: this result mentioned in the conclusions is only supported by evidence in the supplementary materials. I would argue that any conclusion presented here must be supported by material included in the main text.*

Thank you for your comment. We agree that the conclusions should be clearly supported by evidence presented in the main text, rather than relying solely on supplementary materials. We have revised this part to ensure that all statements are directly supported by the main text of the manuscript.

*Line 415: please define "close to river mouth areas"*

Thank you for your comment. "Close to river mouth areas" refers to regions within approximately 20 km from the mouth of a river. We also included this definition in the revised manuscript for clarity.

*Line 417: please specify the contribution of SGD nutrients in percentage to the total, this is the main objective of the manuscript.*

Thank you for your suggestion. This has been stated in the revised manuscript (on lines 449-450) to ensure it aligns with the focus of the study.

*Line 419: I would say it was based on simulations with and without the buoyancy effect, as it is in essence the same model.*

Thank you for highlighting this. We agree with your expression and have made this revision in the revised manuscript.

*Line 426: "the shallow water depth allows for inclusion in the photic zone and thus use by phytoplankton"*

Thank you for your suggestion, which indeed clarifies the expression. We have made this modification in the revised manuscript.

*Line 427: please rephrase to make it clear you are still talking about the SDG-derived nutrients, what distance from the coast you are referring to and why it would be difficult for phytoplankton to use them. Surely you are referring to the lack of dispersal of these nutrients to offshore euphotic areas? That does not make it difficult for plankton to use them, it simply means they have no access to them. Given the objective of this manuscript, listing the relative contributions here (in %) and the area affected (in km) by them should be a priority.*

Thank you for your thoughtful suggestion. We agree with your points and have revised the manuscript following your suggestions. We clarified that the challenge is not the phytoplankton's ability to use the nutrients but rather the nutrients' lack of dispersal to offshore euphotic zones. We also included the relative contributions (4%) and affected areas (~3 km from the shore) of the SGD-derived nutrients in the revised manuscript.

**References**

*Lenhart, H. J., & Große, F. (2018). Assessing the effects of WFD nutrient reductions within an OSPAR frame using trans-boundary nutrient modeling. Frontiers in Marine Science, 5, 447.*

*OSPAR (2013) Distance to target modelling assessment, report 2013-599, ISBN 978-1-909159-32-7, https://www.ospar.org/documents?v=7319*

*Painting, S. J., Van der Molen, J., Parker, E. R., Coughlan, C., Birchenough, S., Bolam, S., Aldridge, J.N., Forster, R.M. & Greenwood, N. (2013). Development of indicators of ecosystem functioning in a temperate shelf sea: a combined fieldwork and modelling approach. Biogeochemistry, 113(1), 237-257.*

*Skogen, M.D., Ji, R., Akimova, A., Daewel, U., Hansen, C., Hjollo, S.S., van Leeuwen, S.M., Maar, M., Macias, D., Mousing, E.A., Almroth-Rosell, E., Sailley, S.F., Spence, M.A., Troost, T., van de Wolfshaar, K. (2021) Disclosing the truth: are models better than observations?, Marine Ecology Progress Series, DOI: 10.3354/meps13574*

Thank you for sharing these references.

---

## Author Comment (AC2)

***Reply to Referee #1:***

*I struggled with the manuscript for many hours trying to figure out what the authors are saying, mainly because the presentation is very poor and very reader-unfriendly. My comments are divided into two categories:*

We apologize for the poor presentation and greatly appreciate your helpful comments and suggestions. Our responses to the comments are as follows. The referee's comments are cited in italics.

*Science:*

*1. I have followed Profs. Guo's and Zhang's work for years and have no doubt about Guo's modeling ability and the high quality data from Zhang's lab. The NPZD model, however, is a very simple one so ground-truthing is important. I failed to notice the comparison of field data and model results although clearly large discrepancies exist. It should also be said whether sea grasses or other plants on the bottom are important.*

Thank you for your valuable feedback.

We have addressed model validation in the supplementary materials (Figure S3), which was mentioned before section 3.1. There are some discrepancies between model outputs and field observations, which likely result from mismatches in the temporal resolution of the observation data (snapshot) and model outputs (monthly mean) as well as their different time periods. However, the spatial distribution and seasonal patterns derived from both observation and modeling are consistent, indicating that the model reasonably captures the major features of ecosystem in the Toyama Bay.

Although seagrasses are present in Toyama Bay, they are not particularly significant in the eastern part of the bay, where submarine groundwater discharge predominantly occurs (Ministry of the Environment, 2008). Therefore, seagrasses and other benthic phytoplankton were not included in our model. We have clarified these aspects in the revised manuscript.

*2. It is not clear why 70m was chosen as the lower limit of the SGD input. How the SGD discharge distributes within this 70m range was also not given.*

The 70-meter depth limit for SGD input was determined based on prior observational studies in Toyama Bay, which indicate that groundwater discharge is primarily concentrated in the nearshore eastern regions of the bay (Hatta and Zhang, 2013). This area is characterized by the steep seabed and the localized nature of SGD, and therefore the zones deeper than 70 m have little groundwater discharge.

Additionally, because the spatial distribution of SGD discharge in this area is not distinctly defined, we assumed a uniform distribution of SGD discharge over all the grid points in this area. We have added this explanation in the revised manuscript.

*3. For coastal modeling work frequently moving boundary is applied. I assume that the land boundary is not moving in this case because the tidal range is very small in the Japan Sea. It should be stated so.*

Thank you for your insightful comments. We agreed and included this clarification in the revised manuscript, stating that a moving boundary is not applied because of the small tidal range in the Japan Sea.

*4. Should explain the huge residual in Fig. 10.*

Thank you for pointing out this issue. We assumed that the initially present nutrients represent the residual term, which are gradually replaced by the modeled nutrient sources over time. If it eventually diminishes to zero, we can judge that the initially present nutrients have been fully replaced the specified external sources of nutrients. If not, we need to define the other external sources of nutrients and make it approach zero. We have added this explanation in the revised manuscript.

*5. Figures 8,9,11,12: not clear whether they are simply comparisons of SGD vs. rivers, or the ocean-side input is considered.*

They are individual simulation results that do not consider ocean-side inputs. The simulations for SGD-derived nutrients only consider inputs from SGD, while those for riverine nutrients solely account for river inputs. We have added annotations to the figure captions for clarity.

*Presentation:*

*1. It is not even clear how much SGD contributes to the study area. It is said at several places that the SGD contributes slightly more nutrients than all rivers combined to the study area. Yet, line 169 says that ...the SGD mean value of DIN loading of 26.7 g/s ...is approximately 20% of total riverine nutrient loading. Only two lines above it is said that the SGD nutrient loading is the same as rivers.*

We are sorry for the misleading description. The value of 20% is for the total riverine nutrient loading into Toyama Bay. "The same as rivers" is for the nearby rivers close to the SGD sites, which represent only a part of rivers along the coast of Toyama Bay. In revision, we will delete the sentence with "is approximately 20% of total riverine nutrient loading".

*2. Figure 10. There are many symbols to use. It is not possible to differentiate red circles from pink ones.*

Thank you for pointing this out. We have revised this figure to improve clarity.

*3. Line 433, Should use positive tone. Instead of saying that "we paid little attention to..." should say that seasonal and short term variations will be the focus of future work.*

Thank you for your suggestions. We have made this change in the revised manuscript.

*4. It would help the readers if the currents are plotted when horizontal distributions of parameters are provided.*

Thanks. We have included a figure showing the currents in the supplementary materials (Figure S4).

*Minor points:*

*1. Line 37 quotes Santos et al., 2021 as summarizing the SGD-related nutrient inputs. Wilson et al. (L&O Letters, 9,4,411,2024) gave a much more comprehensive summary.*

Thank you for your suggestion. We have revised the text to incorporate the more comprehensive summary provided by this literature to ensure the introduction is thorough and up to date.

*2. Figure 1, state that the depth contours are in "m", and put "A, B, C, D" at the proper places.*

Thanks. We have revised this figure following your suggestions.

*3. Figures 4,7 and 8. The X-axis is distance, but from where?*

Thank you for pointing this out. This was our oversight. The distance on the x-axis is measured from the shore and we have added the relevant explanation to the figure captions.

*4. Figure 6 is about phytoplankton but the color bar gives N.*

To maintain consistency, we used the raw model output data (in units of mmol N $\mathrm{m}^{-3}$) to present the phytoplankton concentration.

*5. Figures 7, 8, what is the unit for the color bar?*

The unit for the color bar in Figure 7 is mmol N $\mathrm{m}^{-3}$, while Figure 8 does not have a unit as it represents proportional values. We have provided additional clarification in the figure captions.

*6. Figure 13, areas 1,2,3,4 should be A, B, C, D.*

To differentiate from the labels (A), (B), (C), and (D) used in Figure 1b, we used 1, 2, 3, and 4 here to represent the four different depth areas within the blue line region near the SGD outlet locations shown in Figure 1c.

*7. Line 586, "T"oyama "B"ay; should state (in Japanese).*

Thank you for pointing this out. We have made this change in the revised manuscript.

Reference

Hatta, M. and Zhang, J.: Temporal changes and impacts of submarine fresh groundwater discharge to the coastal environment: A decadal case study in Toyama Bay, Japan, J. Geophys. Res. Ocean., 118, 2610–2622, https://doi.org/10.1002/jgrc.20184, 2013.

Ministry of the Environment: The 7th National Survey on the Natural Environment: Report on Shallow Marine Ecosystems Survey (Seaweed Bed Survey), https://doi.org/https://www.biodic.go.jp/reports2/6th/6_moba19/6_moba19.pdf, 2008.

---

## Referee Report (RR1)

**Second review** of the manuscript *Effects of submarine groundwater on nutrient concentration and primary production in a deep bay of the Japan Sea*, by M. Dong, X. Guo, T. Matsuura, T. Tebakari and J. Zhang, submitted to **Biogeosciences**

**Manuscript overview**

Already given in the first review round.

**Review overview**

Following the first review round the authors made changes to their manuscript in line with the reviewers comments, and have substantially increased the supplementary materials. However, I still miss topics already indicated in the first review round, such as the local scale of the results, the fact that this location has high groundwater discharge and thus that the presented results can serve as a upper limit of the effect of SGD nutrients in a coastal bay area. The difference between the observational period (1931-2001) and the simulated period (2015-2016) is mentioned now, but not further explained. What was the temperature increase in Japanese waters during this period? How did nutrient inputs and population size change? In their reply the authors state "*We have strengthened the comparison of our results with those from other regions in the revised manuscript. Additionally, we also enhanced the presentation of our findings in the abstract and conclusions.*", but I don't see this in the new manuscript much. I also do not see why the authors are so keen on studying this subject, as they do not mention any eutrophication issues in the area or fisheries/aquaculture decline. They use existing work to quantify the reach of this nutrient source, which I find a worthwhile exercise, but more information on why this is important for the region is not given. Is the Toyama Bay area economy focussed on the bay much and thus dependent on its primary production? In short, I still miss some context here. The authors have replied to my comments, but not all of their reply has made it into the manuscript, leaving potential readers with the same questions. This applies to the site description in their reply, the comparison to other areas (the authors include more of a comparison but not the observation that therefore their work can be seen to provide a maximum for SGD nutrient influence) and the zooplankton mortality.

In other replies the authors have failed to substantiate their new text, e.g. in the claim that atmospheric deposition in the area is small compared to other nutrient inputs. They also now mention sewage and industrial loads, but state that these have been added to the riverine loads without further specification (direct discharges are discharges directly into marine waters, so downstream of the last tidal gauge point in a river or directly from the coast). If they know these additional sources, how much do they contribute to the riverine load and where is the reference for them?

I do appreciate the new percentage information, the shifting of figures between the supplementary materials and the main text, and the new figure on the surface current patterns, and think this manuscript is nearly there. Some more points are listed below.

**Recommendation**

Minor revision

**Detailed Comments**

1. Line 19: "*and used by the phytoplankton* for *growth*"

2. Line 129: the term "first class rivers" is still not explained in the text.

3. Lines 167-169: I assume the applied boundary conditions were taking from the larger model for the times specified in the manuscript, and did not consist of climatological values based on the monthly values from the larger model. Is this correct? May be better to state this explicitly.

4. Line 179: how small exactly is the atmospheric deposition? Please provide input (estimates) and their references. Otherwise readers cannot judge whether these depositional values were indeed negligible.

5. Lines 237-242: the authors have added text here to explain the discrepancies between their simulated results (2015-2016) and the observational data (1931-2001), as suggested. But I would still like to see a bit more meat to this bone here. What do observational records show as the T increase in western Japan waters over the total period (1931-2016)? If there was an increase in T then how was this distributed over the year? Surely there are references for this? There is an 1.75 deg.C biased in the T validation. As in general the sea water temperature is easy to get right I strongly suspect the time difference between the obs and the model data, but the authors need to substantiate this.

6. Line 268-272: the authors should specify the different water masses here or refer to a paper that does. What are the S, T characteristics of these different water masses?

7. Line 295: "*changes in detritus* (derived from phytoplankton and zooplankton) *coincide with*"

8. Line 399: I don't see a detritus flux to the sediments in Figure 13, I assume you mean Figure 14.

9. Line 428: I would say there is a difference in the contribution of SGD derived nutrients compared to rivers, and a difference in where in the water column they contribute. But surely their impact on phytoplankton growth is the same as it is for the river-derived nutrients.

10. Line 455: the fact that changes in zooplankton and detritus follow the changes in phytoplankton is not a conclusion from the work presented, but a given when a simple model like an NPZD is used, as is the case here.

11. Line 483: seagrasses are macroalgae and phytoplankton are microalgae. The authors should not refer to seaweed as phytoplankton. And the mention here of limitations to the study is inappropriate: this belongs in the discussion, not the conclusions.

12. Lines 487-489: I disagree. In this manuscript the authors have quantified the contribution of SGD derived nutrients to phytoplankton growth in the bay as 4% of the total. They have also defined the spatial and temporal extent of this contribution, which is limited to the coastal area and occurs in places at depth. Given the fact that it will be difficult to target this nutrient source with management action (in case of eutrophication issues) I would suggest that future work focusses on other issues. For me, the main contribution of this work is the quantification of the SGD nutrient influence to primary production, and its spatial influence sphere. The authors show this is limited in this area, despite the SGD load contribution being large compared to those in other locations. For me, this work is important in that it allows for researchers to disregard this nutrient source (in locations with little or no knowledge of them) as even here, where the source is relatively large, the influence is limited.

---

## Author Response (AR2)

***Reply to Editor:***

***Public justification (visible to the public if the article is accepted and published):***

*Dear Menghong Dong,*

*The expert reviewers have assessed your revised manuscript. While they consider your work suitable for publication, they have highlighted a few outstanding issues. In particular, some revision points raised during the initial review—which you had agreed to address—are still unresolved.*

*One reviewer has provided a detailed report, which I fully support. I would therefore like to ask you to carefully address these comments before I can proceed with the acceptance of your manuscript.*

*Looking forward to your revised submission. Best wishes, Helge*

Dear Dr. Helge,

Thank you for your thoughtful feedback and for giving us the opportunity to revise the manuscript. We sincerely appreciate the reviewers' time and effort in evaluating our work and providing valuable insights.

We acknowledge that some revision points from the initial review still require further clarification, and we have carefully addressed all outstanding issues as highlighted by the reviewer. We have prepared a revised manuscript that incorporates these improvements.

Thank you again for your guidance throughout this reviewing process.

***Reply to Referee:***

***Review overview***

*Following the first review round the authors made changes to their manuscript in line with the reviewers comments, and have substantially increased the supplementary materials. However, I still miss topics already indicated in the first review round, such as the local scale of the results, the fact that this location has high groundwater discharge and thus that the presented results can serve as a upper limit of the effect of SGD nutrients in a coastal bay area. The difference between the observational period (1931-2001) and the simulated period (2015-2016) is mentioned now, but not further explained. What was the temperature increase in Japanese waters during this period? How did nutrient inputs and population size change? In their reply the authors state "We have strengthened the comparison of our results with those from other regions in the revised manuscript. Additionally, we also enhanced the presentation of our findings in the abstract and conclusions.", but I don't see this in the new manuscript much. I also do not see why the authors are so keen on studying this subject, as they do not mention any eutrophication issues in the area or fisheries/aquaculture decline. They use existing work to quantify the reach of this nutrient source, which I find a worthwhile exercise, but more information on why this is important for the region is not given. Is the Toyama Bay area economy focussed on the bay much and thus dependent on its primary production? In short, I still miss some context here. The authors have replied to my comments, but not all of their reply has made it into the manuscript, leaving potential readers with the same questions. This applies to the site description in their reply, the comparison to other areas (the authors include more of a comparison but not the observation that therefore their work can be seen to provide a maximum for SGD nutrient influence) and the zooplankton mortality.*

*In other replies the authors have failed to substantiate their new text, e.g. in the claim that atmospheric deposition in the area is small compared to other nutrient inputs. They also now mention sewage and industrial loads, but state that these have been added to the riverine loads without further specification (direct discharges are discharges directly into marine waters, so downstream of the last tidal gauge point in a river or directly from the coast). If they know these additional sources, how much do they contribute to the riverine load and where is the reference for them?*

*I do appreciate the new percentage information, the shifting of figures between the supplementary materials and the main text, and the new figure on the surface current patterns, and think this manuscript is nearly there. Some more points are listed below.*

Thanks very much for your helpful comments and suggestions. We acknowledge that our revisions in the first round may not have sufficiently addressed the comparison of our results with other regions. Since there are no existing studies quantifying the contribution of submarine groundwater discharge (SGD) derived nutrients in other areas, as you pointed out, under similar conditions, the presented results can serve as an upper limit of the effect of SGD nutrients in a coastal bay area. However, we did not consider benthic phytoplankton or seagrass on the seabed, and the distribution of SGD varies with changes in water depth. Given the complexity of environmental conditions in different marine regions where benthic phytoplankton or macrophytes are important, or where SGD is located at shallower depths, the contribution of nutrients derived from SGD to

phytoplankton growth could be higher. Indeed, in some regions, SGD-derived nutrients can even contribute to eutrophication (Luijendijk et al., 2020). For example, in shallow coastal bays such as Liaodong Bay (Luo et al., 2023) and Zhenzhu Bay (Xu et al., 2024), where the average water depth is less than 30 meters, SGD-derived nutrients may exert a more significant impact and can also lead to eutrophication. Additionally, in areas where seagrass meadows or benthic microalgae are present, SGD can strongly influence the biotic characteristics of seagrass beds (Kantún-Manzano et al., 2018). We have refined our discussion in the revised manuscript (lines 466-476) to reflect these considerations.

Regarding the discrepancies between our simulated results and the observational data, we acknowledge that we did not provide sufficient explanation of the contributing factors. Based on previous observational records, the long-term temperature increase in the western Japan Sea over the past 100 years has been estimated at approximately +1.51°C (JMA, 2024). This warming trend is not uniform throughout the years, and is more in winter and spring temperatures than in summer and autumn (JMA, 2024). Additionally, while the population around Toyama Bay has increased over the decades (Aoki, 2021), the implementation of new wastewater treatment systems has resulted in a 50% reduction in riverine nutrient inputs despite relatively stable river discharge (Katazaki and Zhang, 2021a). Furthermore, climate change-induced shifts from snowfall to rainfall have altered the SGD recharge patterns and chemical composition, leading to lower nutrient concentrations in SGD (Katazaki and Zhang, 2021b). Both the long-term warming trend and the reduction in nutrient inputs contribute to the discrepancies between our model results and the historical observational data. We have mentioned these factors more explicitly in the revised manuscript (lines 251-258) to improve the discussion of model validation.

We appreciate your feedback on the need for additional context regarding the significance of this study for the region. Toyama Bay is known for its high biological productivity and unique oceanographic features, which support commercially important fisheries. While no severe eutrophication issues have been reported (Tsujimoto, 2012), nutrient dynamics play a crucial role in shaping primary production, which directly impacts the ecosystem and fisheries in the bay (NPEC, 2010). Given the recent reduction in nutrient inputs from riverine sources due to improved wastewater treatment (Katazaki and Zhang, 2021a) and the decrease in nutrient concentrations in SGD caused by climate change-induced shifts from snowfall to rainfall in midlatitude Japan (Katazaki and Zhang, 2021b), understanding the role of SGD as a nutrient source is essential for predicting future changes in primary production. Although the Toyama Bay economy is not heavily dependent on the primary production in the bay, fisheries remain an important sector (Bank of Japan Toyama Local Office, 2021). Gaining insight into nutrient supply mechanisms will help assess the potential long-term implications for coastal productivity and resource management. We have enhanced the manuscript by providing more context on the regional importance of SGD-derived nutrients (lines 78-85).

Regarding the exclusion of atmospheric deposition of nutrients, previous studies (Itahashi et al., 2021) estimate that the atmospheric deposition flux of dissolved inorganic nitrogen (DIN) in the study area is approximately 1.2 g/s. This value is significantly smaller than the nutrient fluxes from both SGD and riverine inputs, which is reason why atmospheric deposition was not considered in our model. We have clarified this point in the revised manuscript (lines 190-191).

The riverine loads of nutrients used in our model were from a calculation that estimated the output of nutrients from surface land by including sewage and industrial treatments (Mutsuura et al., 2023).

In this calculation, Mursuura et al. (2023) did not separated the contribution of "direct discharges" from those in the rivers. Therefore, we consider that all the nutrients from surface land were given as riverine loads of nutrients. However, such treatments do not affect the distribution of nutrients in the bay because the nutrients of "direct discharges" flows into the bay along the coast and the riverine nutrients also distribute along the coast. In the realistic bay, these two sources of nutrients should move together.

Our responses to the more detailed comments are as follows. The referee's comments are cited in italics.

**Recommendation**

*Minor revision*

Thank you for the positive evaluation.

**Detailed Comments**

1. *Line 19: "and used by the phytoplankton for growth"*

   Thanks. We have revised this expression on line 19 of the revised manuscript.

2. *Line 129: the term "first class rivers" is still not explained in the text.*

   Thank you for pointing this out. That was our mistake. In Japan, rivers are classified into two main categories based on their significance, scale, and management structure. First class rivers are important for national land conservation and the economy, managed by the national government or prefectural governors under national supervision. Second class rivers are significant for regional public interests and are managed by prefectural governments. We have added this clarification on lines 136-139 of the revised manuscript.

3. *Lines 167-169: I assume the applied boundary conditions were taking from the larger model for the times specified in the manuscript, and did not consist of climatological values based on the monthly values from the larger model. Is this correct? May be better to state this explicitly.*

   Thank you for your comment. We realize that our explanation was not clear in the manuscript. The larger model we used is a climatological model, so the boundary conditions applied in our study were also based on climatological data rather than time-specific values. We have clarified this point explicitly in the revised manuscript (line 178) to avoid any confusion.

4. *Line 179: how small exactly is the atmospheric deposition? Please provide input (estimates) and their references. Otherwise readers cannot judge whether these depositional values were indeed negligible.*

Thank you for your comment. Based on estimates from previous studies (Itahashi et al., 2021), the atmospheric deposition flux of dissolved inorganic nitrogen (DIN) into the study area is approximately 1.2 g/s. This value was derived using reported atmospheric deposition rates in the Japan Sea region and scaled to the surface area of Toyama Bay. We have included the relevant references and detailed calculation in the revised manuscript (lines 190-191) to support our assumption that atmospheric deposition is negligible compared to other nutrient source.

5. *Lines 237-242: the authors have added text here to explain the discrepancies between their simulated results (2015-2016) and the observational data (1931-2001), as suggested. But I would still like to see a bit more meat to this bone here. What do observational records show as the T increase in western Japan waters over the total period (1931-2016)? If there was an increase in T then how was this distributed over the year? Surely there are references for this? There is an 1.75 deg.C biased in the T validation. As in general the sea water temperature is easy to get right I strongly suspect the time difference between the obs and the model data, but the authors need to substantiate this.*

Thank you for your comment. We acknowledge that additional discussion is needed to better explain the temperature discrepancy between our simulated results and the observational data.

Based on previous observational records (JMA, 2024), the long-term temperature increase in the western Japan Sea over the past 100 years has been estimated at approximately +1.51°C. This warming trend is not uniform throughout the year and is larger in winter and spring than in summer and autumn (JMA, 2024). This warming trend may partially account for the 1.75°C bias from model validation. We appreciate your suggestion and have improved this section accordingly in the revised manuscript (lines 251-258).

6. *Line 268-272: the authors should specify the different water masses here or refer to a paper that does. What are the S, T characteristics of these different water masses?*

Thank you for your suggestion. We have included the reference (Hatta et al., 2005) in the revised manuscript (line 291), which specifies the different water masses.

The coastal surface water mass is characterized by low salinity and is influenced by freshwater discharge from rivers into the bay. The Tsushima Warm Current water mass is a warm, saline water mass originating from the Tsushima Current, typically found in the upper layers. In contrast, the Japan Sea Proper water is a cold, high-salinity water mass that dominates the deep layers of the Japan Sea.

7. *Line 295: "changes in detritus (derived from phytoplankton and zooplankton) coincide with"*

Thanks. We have made this change on line 315 of the revised manuscript.

8. *Line 399: I don't see a detritus flux to the sediments in Figure 13, I assume you mean Figure 14.*

Thank you for pointing this out. Yes, this was our mistake. The correct reference should be Figure 14, and we have corrected this in the revised manuscript.

9. *Line 428: I would say there is a difference in the contribution of SGD derived nutrients compared to rivers, and a difference in where in the water column they contribute. But surely their impact on phytoplankton growth is the same as it is for the river-derived nutrients.*

Thank you for your comment. We agree with your statement and have revised the manuscript (lines 447-449) to clarify this point more explicitly.

10. *Line 455: the fact that changes in zooplankton and detritus follow the changes in phytoplankton is not a conclusion from the work presented, but a given when a simple model like an NPZD is used, as is the case here.*

Thank you for your suggestion. Indeed, this correlated change is most likely determined by the characteristics of the NPZD model. We have removed this sentence in the revised manuscript.

11. *Line 483: seagrasses are macroalgae and phytoplankton are microalgae. The authors should not refer to seaweed as phytoplankton. And the mention here of limitations to the study is inappropriate: this belongs in the discussion, not the conclusions.*

Thank you for pointing this out. We acknowledge the mistake in referring to seaweed as phytoplankton and have corrected this in the revised manuscript. Additionally, we also agree that the mention of study limitations should be placed in the discussion section rather than in the conclusions. We have made the necessary adjustments accordingly in the revised manuscript (lines 466-476).

12. *Lines 487-489: I disagree. In this manuscript the authors have quantified the contribution of SGD derived nutrients to phytoplankton growth in the bay as 4% of the total. They have also defined the spatial and temporal extent of this contribution, which is limited to the coastal area and occurs in places at depth. Given the fact that it will be difficult to target this nutrient source with management action (in case of eutrophication issues) I would suggest that future work focusses on other issues. For me, the main contribution of this work is the quantification of the SGD nutrient influence to primary production, and its spatial influence sp here. The authors show this is limited in this area, despite the SGD load contribution being large compared to those in other locations. For me, this work is important in that it allows for researchers to disregard this nutrient source (in locations with little or no knowledge of them) as even here, where the source is relatively large, the influence is limited.*

Thank you for your suggestion. Our original intention was to emphasize that future research should focus on the short-term variations of riverine inputs, as they have a greater impact. However, since the main focus of this manuscript is on the influence of SGD-derived nutrients, we have removed

this part from the text. As you pointed out, considering that SGD-derived nutrient is difficult to be managed and has a relatively minor impact, future work should indeed focus on other issues.